# Carbon footprint of global natural gas supplies to China

Yu Gan [1✉], Hassan M. El-Houjeiri[2], Alhassan Badahdah[2], Zifeng Lu[1], Hao Cai[1], Steven Przesmitzki[3] & Michael Wang[1]

As natural gas demand surges in China, driven by the coal-to-gas switching policy, widespread attention is focused on its impacts on global gas supply-demand rebalance and greenhouse gas (GHG) emissions. Here, for the first time, we estimate well-to-city-gate GHG emissions of gas supplies for China, based on analyses of field-specific characteristics of 104 fields in 15 countries. Results show GHG intensities of supplies from 104 fields vary from 6.2 to 43.3 g $CO_2$eq $MJ^{-1}$. Due to the increase of GHG-intensive gas supplies from Russia, Central Asia, and domestic shale gas fields, the supply-energy-weighted average GHG intensity is projected to increase from 21.7 in 2016 to 23.3 $CO_2$eq $MJ^{-1}$ in 2030, and total well-to-city-gate emissions of gas supplies are estimated to grow by ~3 times. While securing gas supply is a top priority for the Chinese government, decreasing GHG intensity should be considered in meeting its commitment to emission reductions.

[1] Systems Assessment Center, Energy Systems Division, Argonne National Laboratory, Lemont, IL, USA. [2] Climate and Sustainability Group, Aramco Research Center—Detroit, Aramco Services Company, Novi, MI, USA. [3] Strategic Transport Analysis Team, Aramco Research Center—Detroit, Aramco Services Company, Novi, MI, USA. ✉email: ygan@anl.gov

Coal has been China's largest source of energy for several decades[1,2]. However, with the recent concerns about exacerbating air pollution, China has implemented vigorous regulations to promote the coal-to-gas switch[3,4], recently making natural gas the fastest-growing fossil energy in China[2]. In 2017, China's gas supply reached a historic level of 235 billion standard cubic meters (bscm), a dramatic increase of 17% from the 2016 level. The trend will continue according to China's energy plan, and the share of gas in the energy mix is expected to grow from ~6% in 2016 to 15% by 2030 (ref. [1]).

With the limited production capacity of domestic conventional gas[5,6], China has devoted great efforts to obtaining gas from diverse resources[7,8]. China is promoting the exploitation of domestic shale gas resources[9–12], which is estimated as the world's largest technically recoverable shale gas reserves[13]. Besides, imports through international pipelines and liquefied natural gas (LNG) are soaring, making China surpass Japan as the world's largest gas importer in 2018 (refs. [14,15]).

Although the replacement of coal by natural gas is deemed to reduce emissions for China to meet its commitment to lowering carbon intensity by 60–65% from the 2005 level by 2030 (ref. [16]), its effectiveness is affected by the uncertain life-cycle emissions of natural gas[9,17,18]. Approximately 20–50% of the natural gas life-cycle emissions are upstream emissions from well-to city-gate[7,9,17–19], including extraction, processing, and transmission for pipeline gas and additional processes of liquefaction, shipping, storage, and regasification for LNG. The rest of emissions are mainly combustion emissions, which are relatively constant around 50–60 g $CO_2$ eq $MJ^{-1}$ (refs. [18,19]). The difference in well-to-city-gate emissions is the primary reason for the variation of natural gas life-cycle emissions and introduces uncertainties in the climate benefit comparison between gas and coal[9,17,18].

Significant differences in well-to-city-gate emissions occur between gas supplies from diverse sources with different characteristics[20,21]. These differences exist not only across countries or extraction techniques but also among individual fields with distinct geological conditions, raw gas composition, market distances, etc.[20–23]. An engineering-based analysis considering the heterogeneity of individual fields is required to identify underlying drivers of the variability of emissions. The analysis of GHG emissions of natural gas supplies to China at such a granular level has not been done before and will provide insights into emission reductions and clean energy policy-making[24–26].

Here, we analyze well-to-city-gate GHG intensities of gas supplies from 104 fields that produced ~96% of China's supply in 2016. Considering the current and anticipated shares of individual fields, China's natural gas GHG intensity supply curves for 2016 and 2030 are developed. Results show the GHG intensities of the 104 fields range from 6.2 to 43.3 g $CO_2$ eq $MJ^{-1}$. Due to increasing shares of GHG-intensive supplies from Russia, Central Asia, and domestic shale gas fields, the supply-energy-weighted average GHG intensity of China is projected to increase from 21.7 in 2016 to 23.3 g $CO_2$eq $MJ^{-1}$ in 2030. Based on the estimated natural gas GHG intensity supply curves, we discuss potentials for emission reductions and implications for clean energy supply strategies in China.

## Results

**Global natural gas supplies to China.** Table 1 presents China's current and prospective gas supplies from 104 fields in 15 countries. According to statistics, signed import contracts, and domestic production projections, these gas fields represent ~96%, ~95%, and ~89% of China's supply for the years 2016, 2020, and 2030, respectively (see Supplementary Data 1 for details). A significant gap exists between the expected production capacities of these suppliers and the projected demand in China by 2030, for which China is actively seeking additional supply sources. The above shares from the 104 fields could be higher than the current estimates if new contracts are signed with these existing suppliers, e.g., a potential agreement between US and China to expand their current LNG trade.

**GHG intensities of gas supplies from individual fields.** Figure 1 shows the estimated well-to-city-gate GHG intensities of supplies from the 104 gas fields. The supplies are categorized into four types: domestic conventional gas, domestic unconventional gas, international pipeline gas, and overseas LNG (Table 1). Among the four categories, domestic conventional gas has the lowest supply-energy-weighted average GHG intensity of 15.5 g $CO_2$eq $MJ^{-1}$ but the largest within-category heterogeneity, and international pipeline gas has the highest average intensity of 35.9 g $CO_2$eq $MJ^{-1}$. Unless specified, GHG emissions are presented in 100-year global warming potential ($GWP_{100}$)[27]. Results of 20-year GWP ($GWP_{20}$) are presented in Supplementary Fig. 1.

**Domestic conventional gas.** The well-to-city-gate GHG intensities for 34 Chinese domestic conventional gas fields range from 6.2 to 38.9 g $CO_2$eq $MJ^{-1}$. The differences mainly stem from variations in gas transmission and processing (Fig. 1). Transmission emissions vary from 1.3 to 18.1 g $CO_2$eq $MJ^{-1}$, and are primarily determined by the transmission distance for each field (see Supplementary Data 2). In China, the main production areas of natural gas are in the west while the major demands arise from metropolitan areas along the east coast (Supplementary Fig. 2). This uneven spatial distribution of demand and supply leads to the construction of one of the world's longest pipelines, the West–East gas pipeline of ~4000 km[14]. Dina, Kela, and Yingmaili gas fields, located in the western border area, are the main sources that feed into the West–East gas pipeline, and thus have the highest GHG emissions from transmissions. GHG emissions associated with gas processing arise from fuel consumption and fugitive activities during acid gas separation, dehydration, and natural-gas–liquid separation. A high proportion of impurities in raw gas (e.g., $CO_2$, $H_2S$.) would necessitate intensive energy consumption for gas processing. Particularly, the $CO_2$ content, which itself is a GHG, is vented after separation and further increases emissions. For example, the Dongfang and Ledong fields are estimated to have high GHG intensities due to their high $CO_2$ content of >20% in volume (vol%).

Noteworthy, fields with high GHG emissions from gas processing (e.g., Dongfang gas field) tend to have high emissions from extraction as well. This is because gas sources with high impurities require more raw gas extraction to produce an equivalent amount of pipeline quality gas given the feedstock loss from gas processing, thus causing higher GHG emissions. Other factors influencing extraction-associated emissions include the estimated ultimate recovery rate (EUR) per well, well depth, etc. EUR, which is the parameter used to apportion one-time emissions from extraction to lifetime gas production, is estimated based on analyses of field-specific decline curves rather than extrapolations of current productivity rate, reflecting the considerations of future and complete-life production of a gas well. Field-specific values of these parameters are presented in Supplementary Data 2.

**Domestic unconventional gas.** Twenty-five Chinese domestic unconventional gas fields are included in the analysis, covering different types: coal bed methane (CBM, 4 fields), tight gas (17 fields), and shale gas (4 fields). Compared with domestic

**Table 1 Global natural gas supplies to China.**

| Category | Country | Gas fields[a] | % of supply energy[b] | | |
|---|---|---|---|---|---|
| | | | 2016 | 2020 | 2030 |
| Domestic conventional | China | Shuangyushi & Jiulongshan (1), Chunxiao (2), Liwan & Liuhua (3), Anyue-longwangmiao (4), Datianchi (5), Panyu & Huizhou (6), Wolonghe (7), Mahe (8), Kelameili (9), Qingshen (10), Zhongba (11), Sebei (12), Tainan (13), Kekeya (14), Dongping (15), Luojiazhuai (16), Lingshui (17), Donohue (18), Longgang (19), Tieshanpo (20), Bozhong (21), Ya (22), Hetianhe (23), Wenchang (24), Yuanba (25), Puguang (26), Dina2(27), Kela (28), Yingmai7 (29), Tahe (30), Tazhong (31), Ledong (32), Changling & Songnan (33), Dongfang (34) | 32.6 | 30.3 | 20.9 |
| Domestic unconventional | China | Juggar CBM (35), Qingshui CBM (36), Bishuixing CBM (37), Ordos CBM (38), Qiongxi (39), Sulige (40), Guangan (41), Yingtai (42), Hechuan (43), Yulin (44), Zhaotong (45), Daniudi (46), Bajiaochang (47), Changning & Weiyuan (48), Wushenqi (49), Jingbian (50), Yanchang (51), Mizhi (52), Zizhou (53), Shenmu (54), Xinchang (55), Luodai (56), Fuling (57), Dabei (58), Keshen (59) | 29.0 | 28.2 | 28.6 |
| International pipeline | Myanmar | Shwe (60) | 1.9 | 1.2 | 1.9 |
| | Russia | Chayandinskoye (61), Kovyktinskoye (62), Urengoi (63), Nadym (64) | —[c] | 5.6 | 12.6 |
| | Uzbekistan | Karakul (65) | 2.0 | 1.5 | 0.9 |
| | Turkmenistan | Bagtiyarlyk (66), Galkynysh (67) | 14.0 | 8.8 | 12.0 |
| Overseas LNG | Qatar | Qatargas North Field (68) | 3.2 | 3.4 | 2.1 |
| | Oman | Khazzan (69), PDO block 6 (Saih Nihayda, Saih Rawl & Barik, 77) | 0.1 | 0.06 | 0.04 |
| | Russia | South Tambey (70) | —[c] | 1.2 | 0.8 |
| | Australia | Jansz-Io (71), APLNG fields (72)[d], QCLNG fields (76)[e], Gorgon (89) | 7.5 | 8.8 | 5.6 |
| | Papua New Guinea | Hides (73), Angore (74), Juha (75) | 1.4 | 1.1 | 0.7 |
| | Nigeria | Niger Delta (78) | 0.2 | 0.1 | 0.07 |
| | Trinidad&Tobago | Amherstia (79) | 0.1 | 0.06 | 0.04 |
| | Indonesia | Tangguh (80) | 2.2 | 1.3 | 0.9 |
| | Norway | Snohvit (81) | 0.1 | 0.09 | 0.06 |
| | Malaysia | Central Luconia (82) | 1.9 | 1.2 | 0.7 |
| | United States | Gulf of Mexico-offshore (83), Gulf Coast-conventional (84), Central-CBM (85), TX-LA-MS Salt-conventional (86), Central-conventional (87), Gulf Coast-tight (88), Illinois-conventional (90), North Central-conventional (91), Appalachian-conventional (92), Central-tight (93), TX-LA-MS Salt-tight (94), Appalachian-CBM (95), Fort Worth-shale (96), Central-shale (97), Appalachian-tight (98), Illinois-shale (99), Illinois-tight (100), Appalachian-shale (101), West Texas-shale (102), North Central-tight (103), North Central-shale (104) | 0.2 | 1.5 | 1.4 |
| Total | | | 96.3 | 94.5 | 89.4 |

[a]Numbers in parentheses correspond to the gas field numbers in Fig. 1. [b]Detailed shares of supply from individual gas fields and data sources are presented in Supplementary Data 1. [c]— means no gas supply to China for the gas fields in 2016, and the fields are expected to start gas delivery later than 2016. [d]APLNG fields are gas fields of the Australia Pacific LNG project, which include fields of Spring Gully, Talinga, Combabula, Condabri, Peat, Orana, Reedy Creek, Jordan, Ruby Jo, Kenya, Bellevue, Fairview, Arcadia, Roma East, and Ironbark in the Surat and Bowen basin of Queensland Australia. [e]QCLNG fields are gas fields of the Queensland Cutis LNG project, which include fields of Bellevue, Berwyndale, Charlie, Jordan, Kenya, Ruby Jo, and Woleebee Creek.

conventional gas, unconventional gas is estimated with a higher supply-energy-weighted average GHG intensity of 21.4 g $CO_2$eq $MJ^{-1}$, primarily driven by extraction-associated emissions (Fig. 1). The average extraction-associated emissions of Chinese shale, tight, and CBM gas are estimated at 19.1, 14.7, and 9.0 g $CO_2$eq $MJ^{-1}$, respectively, which are significantly higher than that of conventional gas (4.8 g $CO_2$eq $MJ^{-1}$). Compared with conventional gas, additional emissions arise from energy consumption in horizontal drilling and hydraulic fracking to extract tight and shale gas from rock formations with low permeability, and fugitive emissions during the longer flow-back period of well completion and workover. With all types of GHG (i.e., $CO_2$, $CH_4$, $N_2O$) converted to $GWP_{100}$, methane leakages constitute approximately 50–70% of extraction-associated emissions for tight and shale gas. Because methane $GWP_{20}$ is ~3 times the $GWP_{100}$[27], the extraction-associated GHG emissions of unconventional gas increase significantly for $GWP_{20}$ compared to $GWP_{100}$, further amplifying the overall difference between conventional and unconventional gas (Supplementary Fig. 1). Both EUR and initial production rate show significant individual heterogeneities and profoundly influence fugitive emissions from extraction. A higher initial production rate leads to higher fugitive emissions from well completion, while a larger EUR discounts initial episodic emissions, thus reducing the overall GHG

intensity. Chinese CBM fields have slightly higher initial emissions but a significantly lower EUR than those of conventional gas (0.016 versus 0.16 bscm per well), thus higher average GHG intensity associated with gas extraction of CBM. Similar to domestic conventional gas fields, transmission and processing emissions vary significantly according to raw gas compositions and transmission distances for each field.

**International pipeline gas**. GHG intensities of eight international pipeline gas sources range from 11.8 to 41.5 g $CO_2$eq $MJ^{-1}$, with a supply-energy-weighted average of 35.9 g $CO_2$eq $MJ^{-1}$ in 2016, which is the highest among four gas source categories. Transmission-associated emissions increase overall emissions of international pipeline gas due to the extremely lengthy transmission distances (Fig. 1). Methane leakages comprise approximately 50% of the transmission-associated emissions, and because of the higher $GWP_{20}$ of methane than $GWP_{100}$, international pipeline gas is estimated to have even higher transmission-associated emissions if calculated using $GWP_{20}$ (Supplementary Fig. 1). The pipeline leakage rate is a key factor when calculating transmission-associated emissions, yet there is high uncertainty in the estimations of the parameter[19,20,28–32] (Supplementary Fig. 3). Different measurement methods and varied characteristics of the pipeline

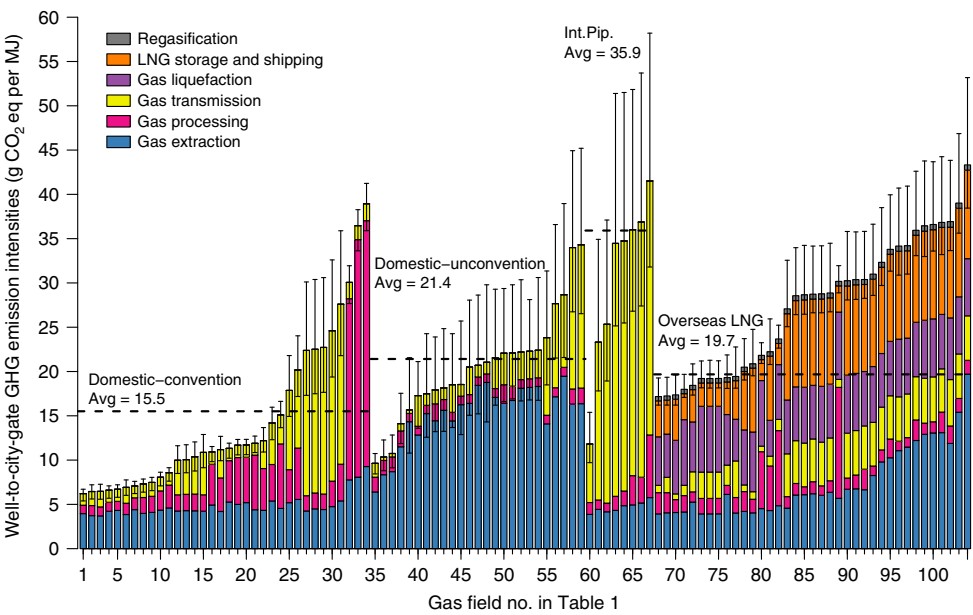

**Fig. 1 Well-to-city-gate GHG intensities of natural gas supplies from individual fields to China using 100-year timeframe global warming potential (GWP$_{100}$).** Different colors show the breakdown of emissions by individual processes. Error bars represent the 90% confidence intervals (CI) of the estimates with Monte Carlo simulation of uncertain parameter inputs (see Supplementary Discussion 1 and 2 for details). Source data are provided in a Source Data file.

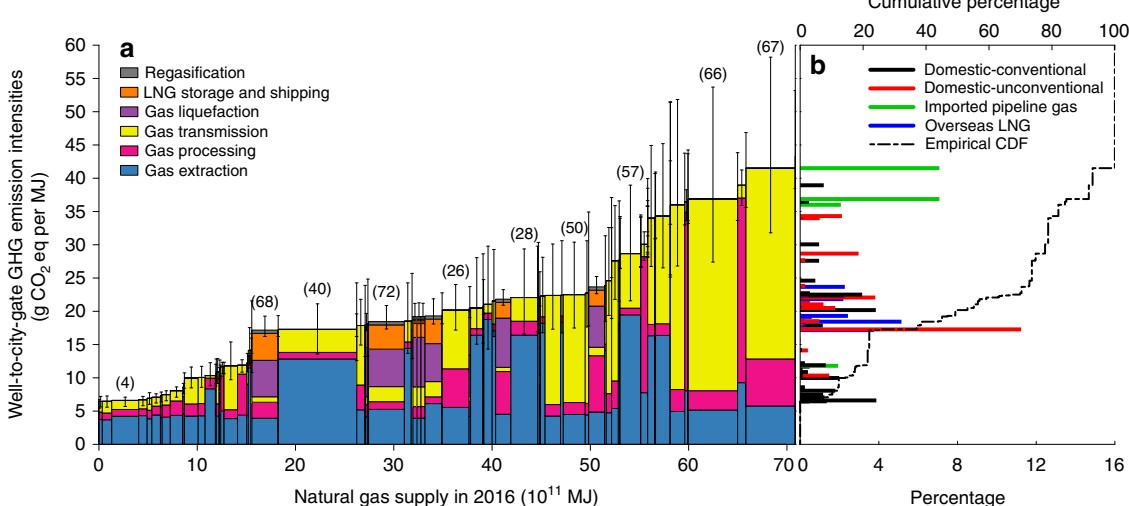

**Fig. 2 Well-to-city-gate GHG intensity supply curve of natural gas for China in 2016. a** The GHG intensity supply curve with emission breakdowns for individual processes. Bars with numbers in parentheses on top are the top 10 gas fields with the largest supply in 2016. The numbers in parentheses are their corresponding gas field numbers in Table 1 and Fig. 1. Error bars represent the 90% CI of the estimates with Monte Carlo simulation of uncertain parameter inputs (see Supplementary Discussion 1 and 2 for details). **b** Empirical probability mass function and empirical cumulative probability function (CDF). Different colors in the empirical probability mass function indicate different categories of gas sources. GHG intensities in the figure are based on GWP$_{100}$. Source data are provided in a Source Data file.

system, such as pipeline age, level of maintenances, and mitigation practices potentially result in variance of pipeline leakage rates (Supplementary Table 1)[28–32]. In the study, the uncertainty range of the pipeline leakage rate is estimated and applied (with other uncertain inputs, see details in Supplementary Table 2) in the Monte Carlo simulations to calculate the uncertainties of results (error bars in Figs. 1 and 2). See Sensitivity Analysis and Uncertainty Analysis in Supplementary Discussion 1 and 2 for details. Supplementary Figs. 4–9 provide sample results for sensitivity and uncertainty analysis.

**Overseas LNG.** GHG intensity for 37 LNG sources ranges from 17.2 to 43.3 g $CO_2$eq $MJ^{-1}$, which is the aggregated result of variations in individual processes. The differences of extraction-associated emissions essentially derive from different extraction techniques (conventional versus horizontal drilling and hydraulic fracking) and production characteristics (e.g., EUR, initial production rate, and well depth). Among all LNG sources, Qatargas North Field has the lowest GHG intensity of extraction (3.9 g $CO_2$eq $MJ^{-1}$) because of its high EUR, while the North Central shale gas from US has the highest extraction emissions (19.7 g

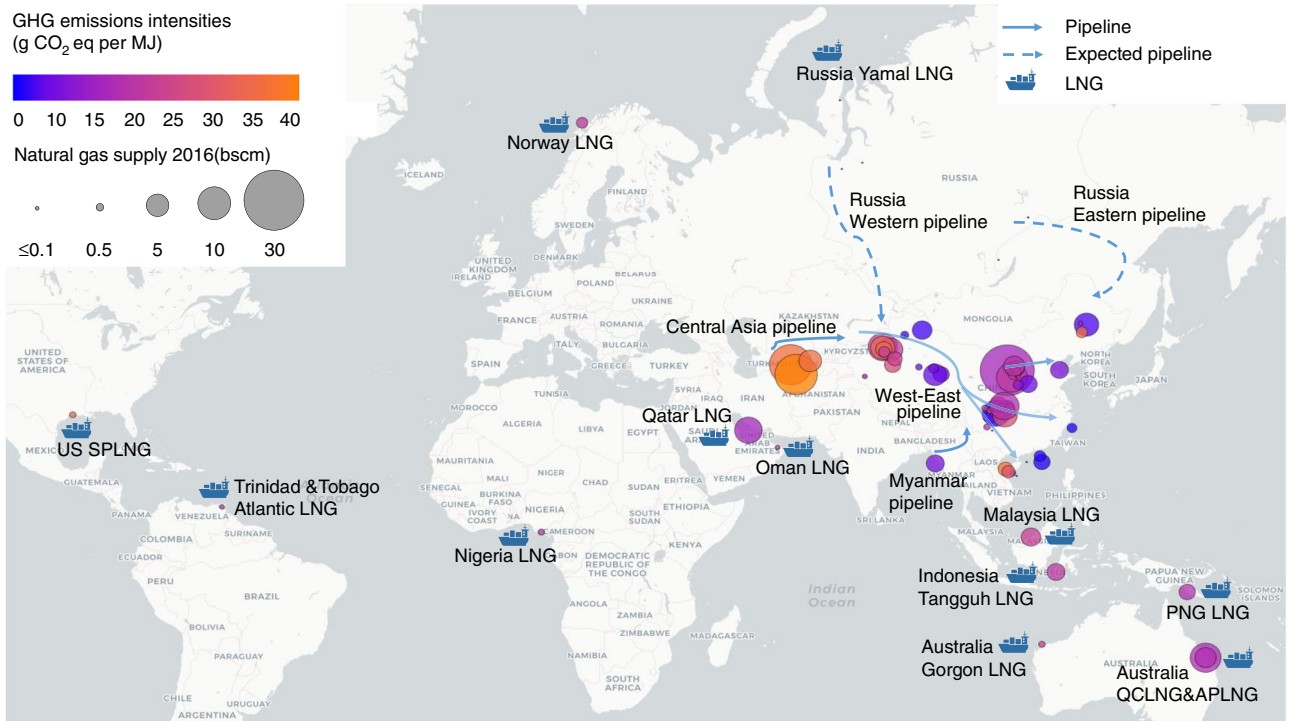

**Fig. 3 Locations of natural gas supplies of China and their corresponding well-to-city-gate GHG intensities in 2016.** Circles in the figure show the location of China's natural gas sources. For overseas liquefied natural gas (LNG), the locations shown are for the LNG terminals. The area of the circle represents the natural gas supply volume in 2016, and the color represents the level of GHG intensity. Natural gas supply with low GHG intensity is colored in blue and gas supply with high GHG intensity is colored in orange. GHG intensities in the figure are based on $GWP_{100}$. Source data are provided in a Source Data file. Map data © OpenStreetMap contributors (https://www.openstreetmap.org/), CC BY-SA.

$CO_2eq\ MJ^{-1}$) due to combined effects of high fugitive emissions of hydraulic fracking and low EUR.

Emissions of gas processing are mainly determined by raw gas compositions. Australia's Gorgon gas field is characterized by the highest average $CO_2$ content (~15 vol%) among all LNG sources and thus the highest processing emissions (12.5 g $CO_2eq\ MJ^{-1}$). A carbon capture and storage (CCS) project was proposed to reduce emissions of the Gorgon project but has been delayed since 2016[33].

Pipeline transmission of gas for LNG supplies includes two parts: transmission from gas fields to LNG plants and transportation from the LNG receiving terminal to city-gate stations or other end-use terminals. Because the latter is assumed the same for all LNG supplies landed in China, the differences in transmission emissions depend solely on the former part. Of all, US Sabine Pass LNG has the longest transmission distance because it collects gas from various fields in the vast area east of the Rocky Mountains[23].

Emissions of liquefaction range from 4.1 to 7.6 g $CO_2eq\ MJ^{-1}$, and the ambient temperature is a key factor driving the differences. Norway's Snohvit LNG plant is the lowest in gas liquefaction because the low ambient temperature facilitates the cryogenic process and improves energy efficiency[34]. Emission of LNG shipping is determined by the distance to China. Shipping LNG from US Sabine Pass to China (~18,000 km distance) is one of the world's longest LNG voyages, while the short LNG shipping distances from the Asian Pacific region are much favored to reduce emissions.

**GHG intensity supply curve and supply map**. Using estimated GHG intensities for the 104 fields (Fig. 1) and their anticipated energy supplies in 2016 (calculated from individual supply volumes and heating values, details in Supplementary Note 1), we plot the

GHG intensity supply curve of natural gas for China in 2016 (Fig. 2 for results in $GWP_{100}$ and Supplementary Fig. 10 for results in $GWP_{20}$). Figure 3 shows the locations of these gas sources and their corresponding well-to-city-gate GHG intensities. The supply-energy-weighted average GHG intensity in 2016 is estimated as 21.7 g $CO_2eq\ MJ^{-1}$, and the values of statistical percentiles (5%, 25%, 50%, 75%, and 95%) are 6.6, 17.2, 19.4, 28.7, and 41.5 g $CO_2eq\ MJ^{-1}$, respectively. Sulige, a domestic tight gas field connected with Shaan-Jing pipeline to provide gas for Beijing, accounts for the largest supply among all fields (11%) and has a lower-than-median GHG intensity of 17.3 g $CO_2eq\ MJ^{-1}$. The Galkynysh and Bagtiyarlyk fields, sources for Central Asia-China pipeline, together account for 14% of supply in 2016, exceeding that of Sulige. GHG intensities of the Galkynysh and Bagtiyarlyk fields are 41.5 and 36.9 g $CO_2eq\ MJ^{-1}$, respectively, higher than the 90th-percentile level. Other major sources include domestic fields of Anyue–longwangmiao (4% of total supply), Puguang (4%), Jingbian (4%), Kela (3%), and Fuling (3%), and international sources of Australia Pacific LNG (4%) and Qatargas LNG (4%). Anyue–longwangmiao is a conventional gas field with a GHG intensity of 6.6 g $CO_2eq\ MJ^{-1}$. Puguang, a conventional gas source of the Sichuan–Shanghai pipeline, has a slightly higher-than-median GHG intensity due to its abnormally high $H_2S$ component. Jingbian, similar to Sulige, is a domestic tight gas field but has a higher GHG intensity (22.1 g $CO_2eq\ MJ^{-1}$) than Sulige owing to its higher $CO_2$ content. Kela is the major source of the West–East gas pipeline and has a higher-than-median GHG intensity of 22.5 g $CO_2eq\ MJ^{-1}$ because of its long transmission distance. Fuling, with a GHG intensity (28.7 g $CO_2eq\ MJ^{-1}$) at the 75th-percentile level, is the largest shale gas field in China. APLNG, which is produced from CBM in Australia, has a lower-than-median GHG intensity of 18.4 g $CO_2eq\ MJ^{-1}$, benefiting from its short shipping distance. GHG intensity of Qatargas LNG, which is the second-largest LNG

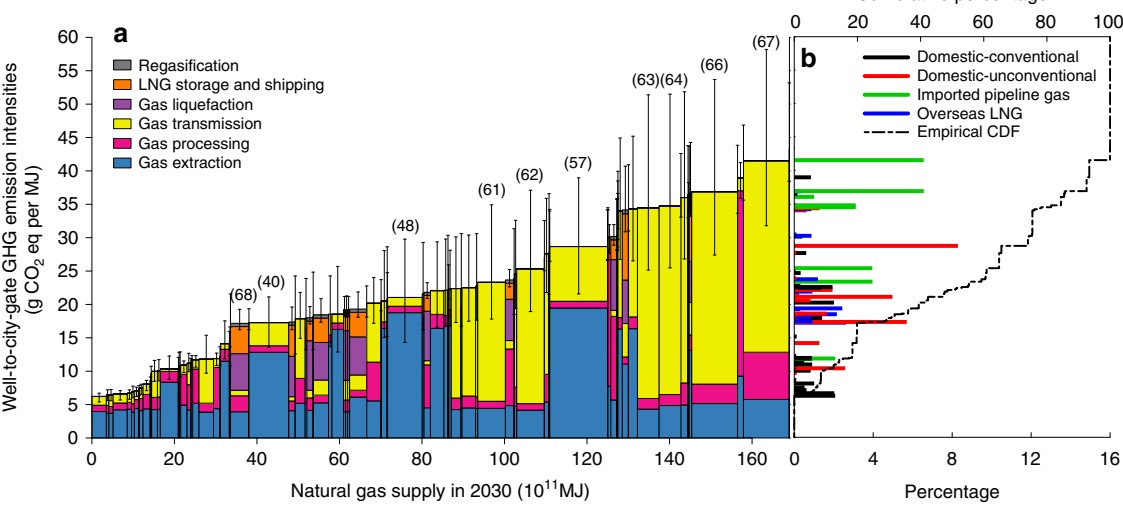

**Fig. 4 Well-to-city-gate GHG intensity supply curve of natural gas for China in 2030. a** The GHG intensity supply curve with emission breakdowns for individual processes. Bars with numbers in parentheses on top are the top 10 gas fields with the largest supply in 2030. The numbers in parentheses are their corresponding gas field number in Table 1 and Fig. 1. Error bars represent the 90% CI of the estimates with Monte Carlo Simulation of uncertain parameter inputs (see Supplementary Discussion 1 and 2 for details). **b** Empirical probability mass function and empirical cumulative probability function (CDF). Different colors in the empirical probability mass function indicate different categories of gas sources. GHG intensities in the figure are based on $GWP_{100}$. Source data are provided in a Source Data file.

source for China (17.2 g $CO_2$eq $MJ^{-1}$), is also lower than the median level because of its high EUR, short pipeline transmission distance, and moderate shipping distance. The above 10 largest gas sources provided more than 50% of China's supply in 2016, and thus have significant impacts on the estimation of GHG intensity supply curve.

**Policy implications through 2030.** The supply-energy-weighted average GHG intensity of 2030 is projected to be 23.3 g $CO_2$eq $MJ^{-1}$ (see Fig. 4 for the GHG intensity supply curve of 2030 in $GWP_{100}$, and Supplementary Fig. 11 for that in $GWP_{20}$). The increasing average GHG intensity of 2030 is caused by the potential growth of GHG-intensive gas supplies, including supplies from Russia's Urengoi and Nadym fields, Turkmenistan's Galkynysh and Bagtiyarlyk, and domestic shale gas from Fulling field, which all have well-to-city-gate GHG intensities higher than the 75th-percentile level at the supply curve. High well-to-city-gate emissions of gas supply would significantly offset the potential climate benefit of China's coal-to-gas switching. Assuming the increase of gas supply in 2016–2030 would replace coal for power generation in China, with an average level of well-to-city-gate GHG intensity, the total GHG emissions reduction benefits are estimated to be 7.4 and 7.8 gigatonne $CO_2$eq in $GWP_{100}$ and $GWP_{20}$, respectively. High GHG intensity at the 80th-percentile level would reduce the climate benefits of the coal-to-gas switching by 1.3 and 2.8 gigatonne $CO_2$eq in $GWP_{100}$ and $GWP_{20}$, respectively. While low GHG intensity at the 20th-percentile level would increase the average climate benefits by 12% and 34% in $GWP_{100}$ and $GWP_{20}$, respectively (see Fig. 5). The variability of well-to-city-gate GHG intensity of gas supplies and its significant effects on climate benefits demonstrates the enormous potentials of further emissions reductions through effective gas resources management and supply chain optimizations. For instance, optimizations of pipeline networks to reduce transmission distances will help reduce emissions and costs. Monitoring and regulating methane leakages of transmission systems can achieve emissions reductions and safety improvements. Applications of CCS to the production of gas fields with exceptionally high $CO_2$ content can effectively reduce GHG emissions.

**Discussion**
The analysis of GHG intensity supply curves provides quantitative information missing in literature for natural gas supply chain GHG management, highlighting challenges and opportunities of emission reductions and clean energy policymaking for expanded natural gas use in China. Results of GHG emissions of individual gas supplies enable Chinese consumers to consider GHG emissions in international and domestic supply contracts to motivate green production. The engineering-based analysis in the study identifies sources and underlying drivers of GHG emissions from gas supplies so that mitigation measures could be considered to achieve China's overall GHG reduction goals.

Although imports are inevitable to bridge the widening gap between domestic gas supply and demand of China, as shown in the study, different import strategies have varied global warming effects. Due to the uneven spatial distribution of demand and supply in China, the eastern coastal metropolitan areas with the largest demand are distant from gas fields in Western Russia and Central Asia (~7,000 km, which is farther than the pipeline distance from Russia to Europe), leading to higher GHG intensity for pipeline gas from these regions. While the eastern coastal metropolitan areas are conveniently located for receiving relatively less-GHG-intensive LNG from Asia-Pacific and the Middle East. This result implies the possibility to adjust the regional distribution of different gas supplies within China to achieve GHG reduction benefits.

The projections of GHG intensity supply curves reflect China's anticipation of domestic production and international trade of natural gas, although high uncertainty exists due to the evolving energy policy and uncertain production status of individual gas fields. For instance, the ambitious target of domestic shale gas exploration is doubtful due to the challenging geology, insufficient technology, and inadequate water resources[9–11]. With a higher average GHG intensity of domestic shale gas compared to overseas LNG, the study questions China's shale gas development regarding its GHG emission-reduction effects.

The present study aims to provide climate-wise choices for China to minimize GHG emissions for its growing natural gas

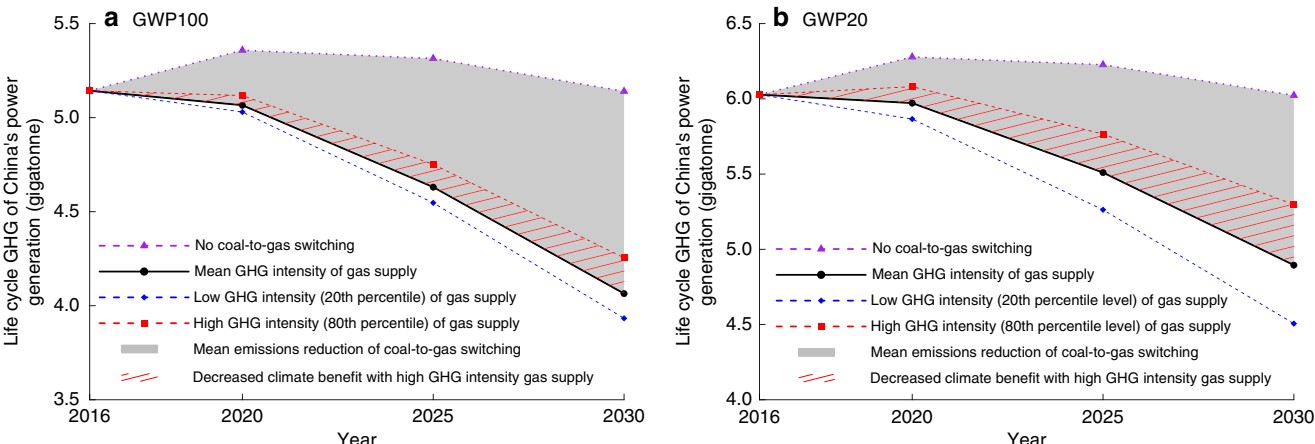

**Fig. 5 Estimation of GHG emissions reduction benefits of China's coal-to-gas switching through 2030. a** Life cycle GHG emissions of China's power generation through 2030 in $GWP_{100}$. **b** Life cycle GHG emissions of China's power generation through 2030 in $GWP_{20}$. Different lines in both panels show the projections of life cycle GHG emissions of China's power generation under difference scenarios: 1. No coal-to-gas switching, 2. Mean GHG intensity of gas supply: all growth of natural gas in 2016–2030 is assumed to replaced coal in the power sector while maintaining the sum of power output from coal and natural gas the same as that in the no coal-to-gas switching scenario. The well-to-city-gate GHG intensity of natural gas in the scenario is assumed at the mean level at the supply curve. 3. Low GHG intensity of gas supply: assume the same volume of natural gas to replace coal as that in the mean GHG intensity of gas supply scenario. The well-to-city-gate GHG intensity of natural gas in the scenario is assumed at the 20th-percentile level. 4. High GHG intensity of gas supply: assume the same volume of natural gas to replace coal as that in the mean GHG intensity of gas supply scenario. The well-to-city-gate GHG intensity of gas supply in the scenario is assumed at the 80th-percentile level. The projections of China's total electricity generation and other energy sources for power generation through 2030 are obtained from the international energy outlook 2019 by US Energy Information Admiration (EIA)[47]. Life cycle GHG emissions intensity of coal and combustion emissions intensity of natural gas are obtained from a China-data-configured GREET®[19]. Source data are provided in a Source Data file.

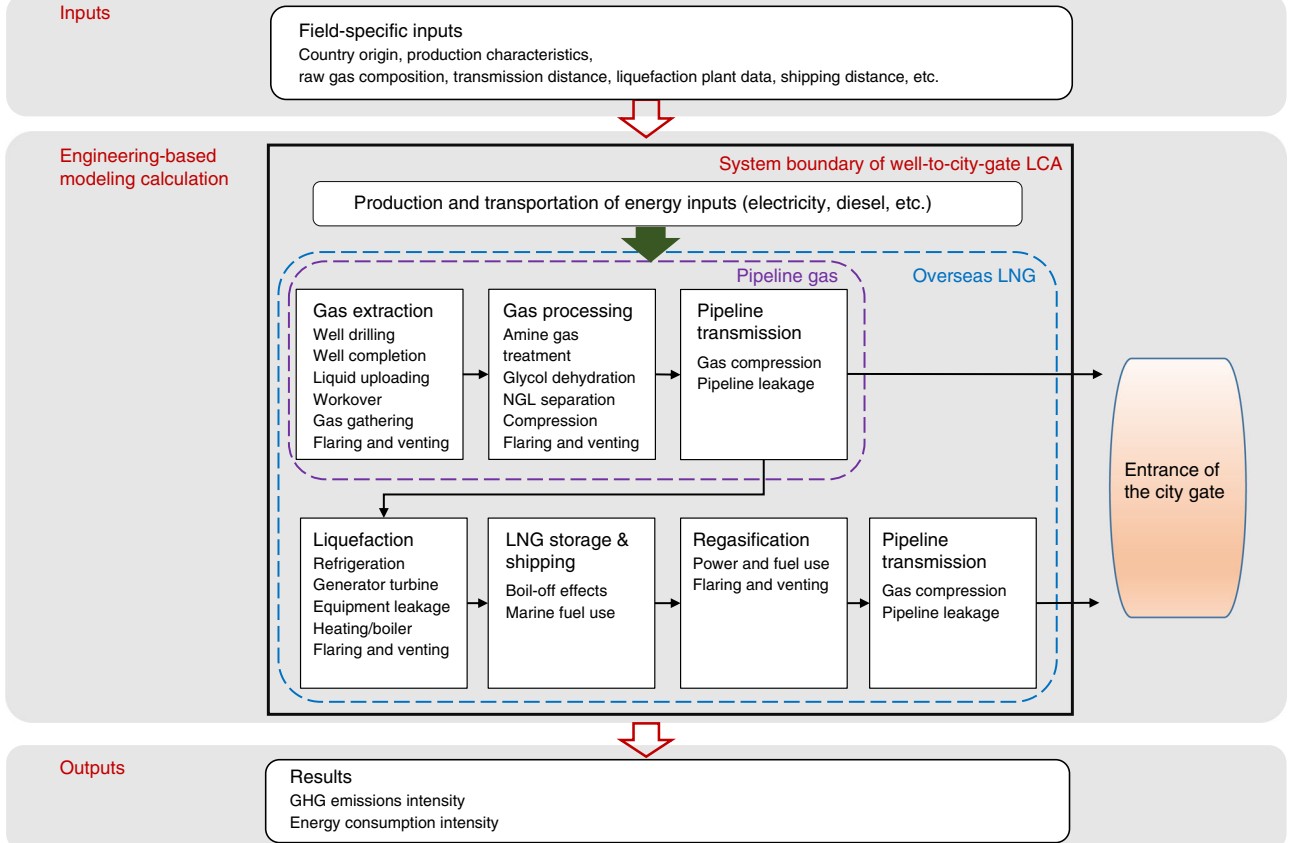

**Fig. 6 System boundary and structure of the natural gas well-to-city-gate LCA model.** Field-specific inputs are fed into the engineering-based LCA model to calculate GHG emissions intensities of gas supply from different gas fields. System boundary of the LCA study is shown in the bold black box in the middle of the figure.

supply. Besides emissions reduction pressures, China's natural gas supply strategies are affected by a variety of factors, including geopolitical relationships, energy security concerns, and economic costs. Economic analysis in Supplementary Discussion 3 compares the cost of supply for different natural gas sources. The supply cost of domestic conventional gas is generally lower than other categories of gas supplies, but its production capacity is limited to meet China's prospective demand. The uncertainty ranges of supply cost for overseas LNG, intentional pipeline gas, and domestic unconventional gas overlap owing to the uncertain production cost of unconventional gas in China and fluctuating global gas price (Supplementary Figs. 12–14 and Supplementary Table 3). More data regarding the production cost and pricing mechanism are required for in-depth economic analysis in the future.

## Methods

**Well-to-city-gate LCA model of natural gas supply in China.** In this study, the well-to-city-gate life-cycle assessment (LCA) model of natural gas supply was compiled by integrating efforts from the National Energy Technology Laboratory's modeling of natural gas extraction, processing, and transmission[20,21] with Argonne National Laboratory's GREET® model for LNG storage, LNG shipping, and offsite generation of fuels and electricity[19], as well as other studies on gas liquefaction[17,23,34–38], LNG regasification[38,39], and model adjustments[7,9,23,26,40–44] (see Supplementary Note 1 for details). The model applies field-specific inputs to empirical engineering modeling of each unit process (e.g., well drilling in the extraction stage, amine gas treatment in the gas processing stage, etc.) to estimate the energy and mass flow through the natural gas supply chain, and thus calculate GHG intensities on a per-MJ gas-delivered basis (see Fig. 6 for system boundary and model structure). Supplementary Tables 4–10 present key assumptions of the LCA model. Local gas distribution after high-pressure pipeline transmission is not included in the system boundary because distribution varies with end-use sectors and therefore is beyond the scope of the study. For instance, large industrial and electricity generation plants receive gas directly from the high-pressure pipeline without distribution while household consumers require additional low-pressure distribution.

**Estimation of transmission distance of pipeline gas.** Transmission distance is a key factor in determining GHG intensities of pipeline-source gas. A detailed analysis was conducted to estimate the transmission distance of each gas field. First, we identified the target consumers for each gas field based on reports from government and enterprises. The pipeline transmission distances between individual gas fields and their corresponding supply destinations were then estimated according to the pipeline system information disclosed by major oil and gas companies in China. Finally, for each gas field, the transmission distances of different supply destinations were multiplied by their corresponding shares of gas demand to calculate the weighted average transmission distance. Because limited data are available for sub-regional gas demand within China, supply destinations were aggregated at the province level. The capital city of each province was treated as the representative destination of the province, except for Xinjiang, for which we took into account the distinct gas supply system in its North and South parts. Urumqi, the capital city of Xinjiang, was chosen as the representative supply destination for North Xinjiang; while Hetian, the midpoint of the South Xinjiang Pipeline, was chosen as the representative destination for South Xinjiang (see Supplementary Data 3 for details).

**Electricity generation mix of gas-origin countries.** The life-cycle GHG intensities of the electricity grids of different gas production countries used in the LCA model were simulated by applying each country's share of energy sources for power generation to the GREET® model[19]. Country-specific shares of energy sources for power generation were obtained from the International Energy Agency's World Energy Statistics[45] (see Supplementary Data 4).

**Reporting summary.** Further information on research design is available in the Nature Research Reporting Summary linked to this article.

## Data availability

All data regarding the parameters used in the natural gas LCA model and data sources are documented in Supplementary Note 1. Field-specific parameters input to the LCA model and the corresponding data sources are presented in Supplementary Data 2. The field-specific data were obtained from various sources including statistics reports, industrial technical papers, and research articles. Less than 10% of the inputs were from commercial dataset (https://www.woodmac.com/our-expertise/capabilities/upstream-oil-and-gas/)[46] to fill the data gap of publicly available data. All other data used in the study are given in Supplementary Data 1, 3 and 4. The source data underlying all figures in the main manuscript and Supplementary Information are provided as a Source Data file.

## Code availability

The Microsoft-Excel-based LCA model which was used to generate the GHG intensities of individual gas fields is available from the corresponding author upon request.

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

## Acknowledgements

This work was supported by the Aramco Services Company under an agreement through US Department of Energy Contract No. DEAC02-06CH11357.

## Author contributions

M.W. and Y.G. conceived of and designed the study. Y.G., H.M.E.-H., A.B. and Z.L. were involved in data gathering, processing, and analysis of different gas fields. The results were interpreted by Y.G. with critical input from H.M.E.-H., A.B., Z.L., H.C., S.P. and M.W. to the discussion. Y.G. led the writing of the paper and all co-authors contributed to the review and revision.

## Competing interests

The authors declare no competing interests.
