## [Peer Review File · Nature Communications]

Reviewers' comments:

Reviewer #1 (Remarks to the Author):

This contribution uses engineering analysis of transmission 'greenhouse gas' (GHG) intensities in order to evaluate the overall well-to-city GHG emissions for gas supplies to China at the present time and then projected out to 2030. This is a valuable contribution to the literature. However, there are a number of points that would help clarify the outputs from this study:-

(1) It would be useful if the authors could confirm that they have applied LCA in accordance with the ISO 14040 series (or some other recognised international standard). This refers, for example, to LCA as being environmental 'Life Cycle Assessment' rather than 'Life Cycle Analysis'.

(2) The contribution would benefit from clarification over the system boundary conditions adopted. Thus, in regard to 'well-to-city' does that refer to the entry gate to the city or the exit gate from the city? The difference concerns operational emissions within the city. This could best be illustrated by way of a schematic diagram depicting the whole supply chain from well-to-city.

(3) There are significant uncertainties around GHG intensities for Chinese gas supplies both at the present time and then projected out to 2030. It would be useful for the authors to indicate the error bars that apply to the parameters they have estimated. One would expect much larger error bars on parameters estimated for 2030 than at the present time.

(4) The authors acknowledge that a major weakness of their study in terms of the projections out to 2030 is the absence of economic analysis. That is clearly difficult in a country like the Peoples Republic of China (PRC) that retains elements of a planned economy.

(5) The present study also relies on data provided by agencies of the Government of the PRC. Some analysts have expressed doubts about the reliability of such data.

Reviewer #2 (Remarks to the Author):

I think the paper addresses a relevant issue, i.e.: what will be the consequences of China's shift from a coal based energy system to natural gas based one? The authors give a good overview of the (projected) Natural Gas sources used by China now and in the near future (2030). An increase in imports from Central Asia (and other regions) as well as the growing role of shale gas in the domestic production will increase the upstream emission per MJ of natural gas. This is a novel finding, which might also be of interest to a wider audience, however the authors can stress this relevance a bit more (see comment 1). The results appear to be well supported by the data; the supplied data in the supplemental excel files seems to be comprehensive and it should be possible to replicate this paper's results. There is one exception: some data points stem from commercial databases; these values are not provided. I do think there are some points that need to be improved, before this paper can be considered for publication, these points are listed below:

Main points to be improved:

1) In its current form the paper focuses quite heavily on the technical aspects, while under-representing its relevance as climate mitigation measure. The main result of the paper is that the upstream emissions of natural gas delivered to China are expected to increase from 21.9 g CO₂eq/MJ in 2016 to 23.7 g CO₂eq/MJ in 2030; which according to the authors might undermine the effectiveness of China's coal-to-gas switch strategy (lines 31-33). It is difficult for the reader to judge how much effect this has on the total GHG emissions in 2030 in China. I would expect that the effect is actually relatively minor and that the benefits of shifting from coal to natural gas greatly outweigh the slightly larger upstream emissions for Natural Gas. The authors can consider

to add a figure with the future trajectory of China's total GHG emissions per year with the current CO₂ intensity of NG and the effect of taking into account the future developments in supply.

2) One of the strong points of this paper is the use of region-specific data with regards to gas composition and pipeline length. Unfortunately, as stated on lines 127-128, the leakage rates from pipelines are generic rather than region-specific. This is a shame because, this is both one of the most critical variables and it is also a variable that is likely to show a high degree of variation across regions. I don't think the +/- 20% used in the sensitivity analysis does justice to this variation. I think the pipeline leakage rate is likely to show a much higher spatial/technological variability than that. This is a flaw that is not easily corrected without gathering additional data. Still I feel that the paper needs to address this issue in a much more comprehensive way before it can be published in Nature Communications. It is crucial to know this because much of the added value of the paper lies in reducing the uncertainty in China's future natural gas footprint. Without having a much better view on the uncertainty in pipeline leakage, it is impossible to judge to which extent the overall uncertainty is reduced.

Stylistic improvements (figures):

3) The color schemes of figures 1, 2 and 4 are a bit harsh on the eye, perhaps pastel versions of the same colors can be used to improve the aesthetics.

4) I also suggest to change the text direction on the vertical axes to improve the readability.

5) Furthermore I would suggest to remove the box around these figures.

6) Please add a little spacing between parts A and B of figures 2 and 4.

Other:

7) I have a strong preference for the use of SI units in the Supplemental Material and the main text, units such as Btu/hphr (Tables S2 and S7) or kg/bbl (Table S2) are unnecessarily complicated. The first one should even be dimensionless since both are measures of energy.

Manuscript ID: NCOMMS-19-28963-T

Carbon footprint of global natural gas supplies to China

Yu Gan, Hassan El-Houjeiri, Alhassan Badahdah, Zifeng Lu, Hao Cai, Steven Przesmitzki, Michael Wang

We sincerely appreciate the editors and reviewers for their comments and suggestions, which have helped us improve the manuscript. Our response to the reviewers is organized as follows: Each comment is repeated (words in blue color, italic), followed by a description of the response (indented) and the changes made in the manuscript (in italic and quotes). We also provide a revised manuscript with revisions in highlights (words in red color). We hope this facilitates the editors and reviewers' evaluation of our response.

Responses to Comments of Reviewer #1

This contribution uses engineering analysis of transmission 'greenhouse gas' (GHG) intensities in order to evaluate the overall well-to-city GHG emissions for gas supplies to China at the present time and then projected out to 2030. This is a valuable contribution to the literature. However, there are a number of points that would help clarify the outputs from this study:

(1) It would be useful if the authors could confirm that they have applied LCA in accordance with the ISO 14040 series (or some other recognised international standard). This refers, for example, to LCA as being environmental 'Life Cycle Assessment' rather than 'Life Cycle Analysis'.

Response: We agree and thank you for the suggestions. We confirm that we have applied the life cycle assessment (LCA) in accordance with the ISO 14040 series. We have revised “life cycle analysis” in the original manuscript to “life cycle assessment”. In the first paragraph of the Methods section:

“In this study, the well-to-city-gate life-cycle assessment (LCA) model of natural gas supply was compiled by integrating efforts...”

The title of Section 1 in the Supplementary Information (SI) I has been changed to:

“Well-to-city-gate life-cycle assessment (LCA) model of natural gas supply”

The following sentence was added in the first paragraph of Section 1.1 of SI I to confirm the application of LCA of the study was in accordance with the ISO 14040 series:

“The life cycle assessment (LCA) of the study is in accordance with the International Organization for Standards (ISO) 14040 series standards.”

(2) The contribution would benefit from clarification over the system boundary conditions adopted. Thus, in regard to 'well-to-city' does that refer to the entry gate to the city or the exit gate from the city? The difference concerns operational emissions within the city. This could best be illustrated by way of a schematic diagram depicting the whole supply chain from well-to-city.

Response: We appreciate and thank you for the suggestions. The “city-gate” refers to the entry gate of the city, and does not include emissions associated with local gas distribution and operations within the city. To address this comment, we have revised Fig.5 (Fig.5 in the original manuscript and Fig. 6 in the revised manuscript) in the methods section to show the natural gas supply chain from extraction at well to entrance of the city-gate and clarify the system boundary of the LCA study.

Fig.5 was revised to:

Fig. 6 System boundary and structure of the natural gas well-to-city-gate LCA model

(3) There are significant uncertainties around GHG intensities for Chinese gas supplies both at the present

time and then projected out to 2030. It would be useful for the authors to indicate the error bars that apply to the parameters they have estimated. One would expect much larger error bars on parameters estimated for 2030 than at the present time.

Response: Thanks very much for the suggestion. The sensitivity analysis in the original manuscript has identified parameters that have significant effects on the final results. In the revised manuscript, we further estimated the probability distribution for those sensitive parameters and then conduct Monte Carlo simulations to calculate the uncertainty of final results.

We added error bars in Fig.1, Fig.2, and Fig.4 to present the uncertainties of the results. We also added a section of uncertainty analysis to present the detailed calculation methods and data sources in SI I.

Fig.1 was revised to:

Fig.1 Well-to-city-gate GHG intensities of natural gas supplies from individual fields to China using 100-year timeframe global warming potential (GWP₁₀₀). Different colors show the breakdown of emissions by individual processes. Error bars represent the 90% confidence intervals (CI) of the estimates with Monte Carlo Simulation of uncertain parameter inputs (see details in SI I). Source data are provided in a Source Data file.

Fig.2 was revised to:

Fig.2 Well-to-city-gate GHG intensity supply curve of natural gas for China in 2016. *a.* GHG intensity supply curve with emission breakdowns for individual processes. Bars with numbers in parentheses on top are the top 10 gas fields with the largest supply in 2016. The numbers in parentheses are their corresponding gas field numbers in Table 1 and Fig. 1. Error bars represent the 90% CI of the estimates with Monte Carlo Simulation of uncertain parameter inputs (see details in SI I). *b.* Empirical probability mass function and empirical cumulative probability function (CDF). Different colors in the empirical probability mass function indicate different categories of gas sources. GHG intensities in the figure are based on GWP₁₀₀. Source data are provided in a Source Data file.

Fig.4 was revised to:

Fig.4 Well-to-city-gate GHG intensity supply curve of natural gas for China in 2030. *a.* GHG intensity supply curve with emission breakdowns for individual processes. Bars with numbers in parentheses on top are the top 10 gas fields with the largest supply in 2030. The numbers in parentheses are their corresponding gas field number in Table 1 and Fig. 1. Error bars represent the 90% CI of the estimates with Monte Carlo Simulation of uncertain parameter inputs (see details in SI I). *b.* Empirical probability mass function and empirical cumulative probability function (CDF). Different colors in the empirical probability mass function indicate different categories of gas sources. GHG intensities in the figure are based on GWP₁₀₀. Source data are provided in a Source Data file.

A section of uncertainty analysis was added in the SI I:

“3. Uncertainty analysis

The above sensitivity analysis has identified parameters that have significant effects on the final results. In this section, we will estimate the probability distributions for those sensitive parameters and then conduct Monte Carlo simulations to calculate the uncertainties of final results.

3.1 Uncertainty of pipeline leakage rate

Leakage rate of pipeline transmission is a key factor to determine the transmission-associated and the overall emissions, especially for gas sourced from long-distance international pipelines. Enormous research efforts have been done on the measurements of pipeline system leakages since the 1990s¹⁻¹⁹. Yet there is still high uncertainty and debate on this issue^{6-8,20}.

To have a clear understanding of the uncertainty of pipeline leakage rate and its impacts on our final results, here, we conduct a literature review on the pipeline leakage studies and incorporate the uncertainty of the pipeline leakage rate into the Monte Carlo simulation to generate robust results.

The majority of studies on pipeline transmission leakages were centered on the U.S. and Russia, which are the two largest natural gas producers that account for ~40% of gas production worldwide⁷. Pipeline leakage rates are also measured and reported in some European countries, such as United Kingdom, Germany and the Netherlands^{5,9}, but since China does not import natural gas from these European countries, studies in those countries will not be discussed here. There is no publicly available report on the leakage rate of China’s pipeline system.

Fig. S5 summarizes literature estimates of the pipeline leakage rate for the U.S. and Russia. Noteworthy, we have converted the percentage leakage rate of throughput to the leakage rate on per kilometer (km) of transport distance basis. The lengths of the U.S and Russian pipeline system that used for conversions are obtained from the literature. Studies of Russia’s pipeline system leaks are focused on Russia’s export corridors to Western European. The length of the export corridors is ~3000 km within the Russian border (the distance of Northern Corridor is 3075 km while the Central Corridor is 3376 km)^{2,11}. The average transmission distance of the U.S natural gas system was estimated at ~1000 km²¹⁻²⁴. The much longer transmission distance of Russia’s natural gas system leads to its higher leakage rate in terms of percentage throughput than that of the U.S. system in literature^{7,10,11}. Yet on the per km basis, pipeline leakage rates of the two systems are comparable and their uncertainty ranges overlap mostly, as shown in Fig. S5.

Fig. S5 Literature estimates of leakage rates of the U.S. and Russia pipeline system.

Data sources: Zimmerle et al. 2015¹², Logan et al. 2012¹³, Laurenzi et al. 2013¹⁴, Alvarez et al. 2018¹⁵, USEPA 2013²⁵, USEPA 2014²⁶, USEPA 2015²⁷, USEPA 2016²⁸, USEPA 2017²⁹, USEPA 2018³⁰, USEPA 2019³¹, Dedikov et al. 1999², Lelieveld et al.^{10,11}, Russia NIR 2012³², Russia NIR 2013³³, Russia NIR 2014³⁴, Russia NIR 2015³⁵, Russia NIR 2016³⁶, Rosstat 2015³⁷

The difference in the estimations of pipeline leakage rate is caused by many reasons. Firstly, varied measurement approaches would result in different estimates²⁰. There are generally two types of measurement approaches: the top-down and the bottom-up approaches. Top-down studies use aircraft, tower, remote sensing, etc. to measure the concentrations of CH₄ or other volatile organic compounds (VOCs) in the atmosphere^{7,15,20}. The atmospheric observations are then attributed to different sources, including multiple anthropogenic sources (such as waste landfill, production of ruminant animals, oil, coal and natural gas production, etc) and natural sources (such as natural wetland, natural geologic seeps, etc.)^{20,31}. The sources attribution is the key challenge for the top-down study and it will introduce significant uncertainties since assumptions about the activity levels of different sources are highly uncertain²⁰. Bottom-up studies measure emissions of sampling devices or facilities, and then multiple the emission factors by device counts or activity factors to make emissions inventories^{15,20,25-31}. According to the literature review by Brandt et al, inventories generated from bottom-up studies, such as the U.S. Environmental Protection Agency (EPA) GHG inventory (GHGI), systematically reported methane emissions lower than top-down measurements²⁰. Limited sample sizes and representatives, uncaptured “superemitters”, and incomplete statistics of activities and devices are responsible for the potential underestimates of bottom-up studies^{7,20}. Fig. S5 includes estimates from both top-down and bottom-up studies, which should have captured the uncertainties from different measurement approaches.

Secondly, characteristics of the pipeline systems, such as pipeline age, operating pressure, level of maintenances and mitigation practices, would affect the pipeline leakage rates^{7,10,18,19,38}. Previous researches revealed that old compressor stations might lead to higher leaks, and pipeline system with higher operating pressures has the tendency to increase leaks, while well maintenance and mitigation measures can significantly reduce transmission leakages^{7,10,18,19,38}.

Table S9 compares the characteristics of China's pipeline system with those of the U.S. and Russia to look for clues about the level of China's transmission leakage rate. As shown in Fig. S5, compared to U.S. and Russia, China has a much younger pipeline system which might help reduce leaks while a relatively higher operating pressure (10 MPa for the major gas pipelines such as West-East gas pipeline and Central-Asia gas pipeline), which, on the other hand, would potentially increase leaks. Methane emissions are not regulated and reported in China and the country does not actively promote emissions mitigation practices in its pipeline system operations. In contrast, U.S. and Russia have promoted and deployed multiple methane emission mitigation measurements, such as forward line pumping, corrosion repair, replacement of high-bleed pneumatic devices, targeted inspection, etc, which are considered to help significantly reduce transmission leaks^{7,16-18}. Kiefner and Rosenfeld studied pipeline incidents (including leaking seals and corrossions that are related to gas leaks) reported to U.S. Pipeline and Hazardous Material Safety Administration (PHMSA) during 2002~2009 and found that 85% of the pipeline incidents are in irrespective of the pipeline age, with just 15% of the incidents related in some way to the pipeline age, and they concluded that periodically assessment, timely repairs with mitigation efforts can ensure aged pipeline's continued fitness for service³⁸.

After carefully comparing the characteristics of the Chinese pipeline system to those of the U.S. and Russia, we are not expecting the Chinese pipeline leakage rate to be significantly higher or lower than that of the U.S. or Russia. Literature estimates of pipeline leakage rates for the U.S. and Russia have wide uncertainty ranges, which should be large enough to capture the possible value for the Chinese pipeline leakage rate. According to Lelieveld et al, the true value of leakage rate for Russia's gas export transmission system must be lower than their upper estimates of 1.6% (~7.1E-06 kg/kg-km, the highest estimate for Russia in Fig. S5), because the upper estimation is calculated based on "worst-case assumptions" in numerous areas¹⁰. Balcombe et al believed that estimates of the leakage rate of pipeline transmission (exclude leakages from production and processing) above 1.6% are results of outdated data or flawed estimation methods⁷.

In the next section, we will apply the uncertainty estimates of Russia's pipeline leakage rate (which has a larger uncertainty range than that of U.S, as shown in Fig. S5) to that of the Chinese pipeline system and the connected international pipeline system, such as the Central Asia-China pipeline, and conduct Monte Carlo simulation for uncertainty analysis.

Table S9 Comparison of characteristics of natural gas pipeline system in different countries

	Year of Pipeline installed	Operating pressure	Maintenance and mitigation practice
China's pipeline system	2000s	6.3~12 MPa	No regulation and measurements on methane emissions
U.S pipeline system	Median at 1960s	3.5~10 MPa	Well-maintained with deployment of emissions mitigation practices
Russian gas export pipeline system	Median at 1980s	7.5 Mpa	Monitored leaks, promoted and deployed emissions mitigation practices

Data sources: ANL 2007³⁹, INGAA³⁸, Lelieveld et al^{10,11}, Balcombe et al⁷.

3.2 Uncertainty of other sensitive parameters

Besides the transmission leakage rate, variations of some other parameters also have significant impacts on our LCA results. According to previous sensitivity analysis, parameters with elasticity greater than 0.1 (which means the final results would change more than 1% with 10% change in the parameter) include: CO₂ content in raw gas, average production rate, initial production rate, EUR, pipeline transport distance, flaring rate at extraction stage, well completion flowback period, compressor efficiency, energy conversion factor of NG engine prime mover, energy conversion factor of NG turbine prime mover, flaring efficiency, energy intensity rate of pipeline transmission, LNG boil-off rate, recovery rate of boil-off gas, ocean tanker average speed. Among these parameters, CO₂ content in raw gas, average production rate, initial production rate, EUR, pipeline transport distance are field-specific inputs, of which the individual variations have been analyzed thoroughly. Here we assumed ±10% variation for these field-specific inputs, except for the EUR, of which ±50% of variations have been assumed because of the relatively high uncertain feature of the parameter. For other sensitive parameters, we estimate their uncertain range through literature reviews, as shown in Table S10.

Table S10 Probability distributions of parameters used in Monte Carlo simulation (field-specific parameters excluded)

Parameter	Uncertainty range	Data sources
Flaring rate conventional gas (%)	Triangular(41,51,61) ^a	NETL ^{21,22}
Flaring rate unconventional gas (%)	Triangular(12,15,18)	NETL ^{21,22}
Flowback period shale/tight gas (days)	Triangular(3,7,10)	NETL ^{21,22}
Compressor efficiency (%)	Triangular(70,75,85)	Simpson ⁴⁰
Energy conversion factor of NG engine prime mover	Triangular(2.38,3.11,3.27)	OPGEE ⁴¹
Energy conversion factor of NG engine prime mover	Triangular(2.76,3.62,4.08)	OPGEE ⁴¹
Flaring efficiency (%)	Triangular(80, 98,100)	J Willis et al ⁴²
Energy intensity rate of pipeline transmission (MJ/tonne-km)	Triangular(0.29,1.19,1.62)	Müller-Syring ⁹ , NETL ^{21,22} , GREET
Pipeline leakage rate (kg/kg-km)	Uniform(2.2E-06, 5.84E-06) for U.S. pipeline system Triangular(1.4E-06, 3.0E-06, 7.1E-06) for others	Zimmerle et al. 2015 ¹² , Logan et al. 2012 ¹³ , Laurenzi et al. 2013 ¹⁴ , Alvarez et al. 2018 ¹⁵ , USEPA 2013~2019 ²⁵⁻³¹ , Dedikov et al. 1999 ² , Lelieveld et al. ^{10,11} , Russia NIR 2012~2016 ³²⁻³⁶ , Rosstat 2015 ³⁷
LNG boil-off rate (%)	Triangular(0.08,0.10,0.15)	Dobrota et al ⁴³ , Głomski and Michalski ⁴⁴ , Sedlaczek ⁴⁵
Recovery rate of boil-off gas (%)	Triangular(70,80,90)	Kwak et al ⁴⁶
Ocean tanker average speed (km/hour)	Triangular(8.8,12.5,14.1)	IMO ^{47,48}

a Triangular(*a,b,c*) means the value input in the model is a triangular distribution, with the lower bound of *a*, upper bound of *c*, and most likely value of *b*.

3.3 Monte Carlo simulation

With the probability distributions of these sensitive parameters, we conducted 5000 times Monte Carlo simulations to calculate the uncertainty ranges of the final results. Well-to-city-gate GHG intensities with 90% confidence interval (CI) for each gas fields are presented as error bars in Fig.1, Fig.2, Fig. 4 in the main manuscript and in Fig.S11-S15. Detailed numbers can be found in SI III.

Fig. S6 and Fig. S7 show the probability density function (PDF) of the well-to-city-gate GHG intensity of gas supply from Galkynysh field.

Fig. S6 Probability density function (PDF) of well-to-city-gate GHG emissions in GWP100 for gas supply from Galkynysh gas field. The PDF is generated based on results of 5000 times of Monte Carlo simulations. GHG emissions in the figure are characterized with 100 years global warming potential (GWP_{100}) factors.

Fig. S7 Probability density function (PDF) of well-to-city-gate GHG emissions in GWP20 for gas supply from Galkynysh gas field. The PDF is generated based on results of 5000 times of Monte Carlo simulations. GHG emissions in the figure are characterized with 20 years global warming potential (GWP_{20}) factors.

(4) The authors acknowledge that a major weakness of their study in terms of the projections out to 2030 is the absence of economic analysis. That is clearly difficult in a country like the Peoples Republic of China (PRC) that retains elements of a planned economy.

Response: Thanks very much for the comment. In the study, we projected the possible GHG intensity natural

gas supply curve for 2030 based on China's current anticipation of domestic production, signed and expected import contracts. Through the analysis of the projected supply curve, we aimed to identify the potential hotspots and underlying driving force for the GHG emissions and thus provide suggestions on emissions reductions for decision-making and natural gas industry development in China. Besides emissions reduction pressures, China's natural gas supply strategies are affected by a variety of factors, including geopolitical relationships, energy security concerns, and economic factors. Analyses of these factors were beyond the original research scope of the study. In the revised manuscript, a section of economic analysis was added in SI I to provide some discussions on the economic aspect:

“4. Economic Analysis

The present study focuses on providing climate-wise choices for China to minimize GHG emissions for its growing natural gas supply. One important question remains regarding the economic cost and its effects on China's choice of global natural gas consumption. In the section, we compare the cost of supply for different gas sources and thus provide further insight into the economic implications of China's natural gas supply policy.

4.1 Overseas LNG

The cost of overseas LNG supply includes the import price and other additional costs, such as the cost of regasification^{49,50}. The LNG import price accounts for more than 95% of the total supply cost of overseas LNG^{49,50}. The LNG import price is positively related to the global oil price and also affected by the market supply and demand conditions⁵¹. The growth of Spot LNG price can break through the trend of global oil price when there are strong demand and tight supply in the LNG market. Impacted by the fluctuating oil price and uncertain market conditions, LNG import price varies in a wide range. As shown in Fig. S8, from January to October 2019, China's LNG import price varied between 280~420 us dollar (USD) per thousand cubic meters (kcm, on the basis of gas volume after regasification)⁵².

Fig. S8 Delivered ex-ship (DES) price of LNG imports for China in 2019.
Data source: Shanghai Petroleum and Gas Exchange (SHPGX)⁵²

4.2 International pipeline gas

The import price of international pipeline gas is determined by the pricing formula in the long-term contract, which has never been disclosed by China National Petroleum Corporation. According to the published statistics of the import prices of international pipeline gas in recent years^{53,54}, the price of international pipeline gas has positive relationship with the global oil price and an extra price increase poses when there are tight supply and strong demand. Generally speaking, the average import price of pipeline gas is 20%~30% lower than that of overseas LNG during the same period^{51,53,54}. However, since the international pipeline gas is mainly imported from the western border that is distant from the eastern coastal metropolitan areas, the extra cost of long-distance pipeline transmission significantly increases the total cost of supply. According to the study by Rioux et al⁵⁰, after including the cost of delivering gas across ~4000 km, the total cost of pipeline gas imports from Central Asia to Shanghai is similar to the coastal LNG import price. Table S11 shows the import price of international pipeline gas in 2018 and the estimations of gas delivering cost within China for different import sources.

Table S11 Import price of intentional pipeline gas in 2018 and estimates of transmission costs

Country	Import price 2018 (USD/kcm)	Cost of gas transmission within China (USD/kcm)							Average
		To Guangdong	To Fujian	To Shanghai	To Zhejiang	To Guangxi	To Henan	To Yunnan	
Turkmenistan	393	134	140	138	137	148	103	--	128
Uzbekistan	374								
Kazakhstan	358								
Myanmar	603	--	--	--	--	46	--	16	30

Data sources: International Trade Centre (ITC)⁵³, General Administration of Customs of China⁵⁴, Rioux et al⁵⁰, Zhang et al⁴⁹.

4.3 Domestic gas

Supply cost for domestic gas is relatively low but its production capacity is limited and far below China's prospective gas demand in the future^{49,55,56}. The average production cost of domestic conventional gas varies between 20~110 USD/kcm in different provinces^{49,50}, as shown in Fig S9. Even with the cost of transmission, the total cost of domestic conventional gas supply is still much lower than the average import price of LNG in recent years^{53,54}. China is now actively promoting the exploration of unconventional gas, such as shale gas in Sichuan Basin⁵⁷⁻⁶⁰. The production cost of shale gas in literature is estimated to be 130~400 USD/kcm^{49,50,61}. With the cost of transmission, the total cost of domestic shale gas can be close to the gas import price.

Fig. S9 Production cost of conventional gas in different provinces of China.
Data sources: Rioux et al⁵⁰, Zhang et al⁴⁹.

Fig S10 compares the supply costs of different sources of gas supply to Shanghai. Due to the lack of public data from natural gas production sector and the involving international gas market, high uncertainty exists in the estimations of natural gas supply costs.

The comparison of supply costs for different categories of gas sources varies for different regions of China owing to the different production cost of domestic gas and relatively transport distance of gas imports. More in-depth economic analysis is required regarding the heterogeneity between different gas sources and different regions of gas consumption, which can be the focus for our future studies.

Fig. S10 Estimated range of cost for different categories of gas supply to Shanghai

Data sources: Rioux et al⁵⁰, Zhang et al⁴⁹, International Trade Centre (ITC)⁵³ General Administration of Customs of China⁵⁴, Shanghai Petroleum and Gas Exchange (SHPGX)⁵², General Electric(GE)⁶¹.

”

The following paragraph was added at the end of the Conclusion section in the main manuscript:

“The present study aims to provide climate-wise choices for China to minimize GHG emissions for its growing natural gas supply. Besides emissions reduction pressures, China’s natural gas supply strategies are affected by a variety of factors, including geopolitical relationships, energy security concerns, and economic costs. Section Economic Analysis in SI I compares the cost of supply for different natural gas sources. The supply cost of domestic conventional gas is generally lower than other categories of gas supplies, but its production capacity is limited to meet China’s prospective demand. The uncertainty ranges of supply cost for overseas LNG, intentional pipeline gas and domestic unconventional gas overlap owing to the uncertain production cost of unconventional gas and fluctuating global gas price. More data regarding the production cost and pricing mechanism is required for in-depth economic analysis in the future.”

(5) The present study also relies on data provided by agencies of the Government of the PRC. Some analysts have expressed doubts about the reliability of such data.

Response: Thanks very much for the comment. In the study, the final results of China’s natural gas GHG intensity supply curve are developed based on the estimations of GHG intensities for different gas fields and the gas supply volume of each field.

For the GHG intensities of different gas fields, the estimates are relied on an engineering-based LCA model, of which most of the technical parameters are obtained from peer-review paper, academic researches, industry reports and database. Pipeline transmission distance is the only parameter that was estimated with data from the Chinese government. The government's statistic data of natural gas consumption in different provinces were used as the weighting factor to calculate the average transmission distance to each province. According to IPCC guidelines, the uncertainty range of energy activities is $\pm 5\%$ for developed countries and $\pm 10\%$ percent for the less developed energy systems⁶². In our study, a $\pm 10\%$ uncertainty has been assumed for the parameter of pipeline transmissions distance in the Monte Carlo simulations to generate robust results.

For the values of gas supply for each field, the government's supply data is not the major data source. The primary sources are information disclosed by oil companies, oil company affiliated research institutions, industry surveys, and public databases. The government's energy statistic data was compared to the sum of gas supply from individual fields for cross-validation. We also used other data sources such as BP Statistical Review⁶³, International Energy Agency's World Energy Statistics⁶⁴ for cross-validation. The gap in the total gas supply between these different data sources is 2~3%. According to Guan et al, the low quality of data reported by small enterprises is an important reason for potential errors in the energy statistics of China. China's natural gas production is controlled and reported by three giant oil enterprises, which has avoided this problem⁶⁵. Different conversion factors (which are used to convert the physical units to energy units) might also introduce potential errors in the energy statistics^{65,66}. In this study, we only used the natural gas supply data in physical units from energy statistics and computed the conversion factors according to the gas composition of different fields to reduce the potential errors. After all, our study is not aimed at accounting the total GHG emissions for the natural gas sector but to analyze the heterogeneity of different gas sources and identify the potentials for GHG emissions reductions. The potential errors in the value of total gas supply of China would not significantly affect the robustness of the results in the study.

Responses to Comments of Reviewer #2

I think the paper addresses a relevant issue, i.e.: what will be the consequences of China's shift from a coal based energy system to natural gas based one? The authors give a good overview of the (projected) Natural Gas sources used by China now and in the near future (2030). An increase in imports from Central Asia (and other regions) as well as the growing role of shale gas in the domestic production will increase the upstream emission per MJ of natural gas. This is a novel finding, which might also be of interest to a wider audience, however the authors can stress this relevance a bit more (see comment 1). The results appear to be well supported by the data; the supplied data in the supplemental excel files seems to be comprehensive and it should be possible to replicate this paper's results. There is one exception: some data points stem from commercial databases; these

values are not provided. I do think there are some points that need to be improved, before this paper can be considered for publication, these points are listed below:

Main points to be improved:

1) In its current form the paper focuses quite heavily on the technical aspects, while under-representing its relevance as climate mitigation measure. The main result of the paper is that the upstream emissions of natural gas delivered to China are expected to increase from 21.9 g CO₂eq/MJ in 2016 to 23.7 g CO₂eq/MJ in 2030; which according to the authors might undermine the effectiveness of China's coal-to-gas switch strategy (lines 31-33). It is difficult for the reader to judge how much effect this has on the total GHG emissions in 2030 in China. I would expect that the effect is actually relatively minor and that the benefits of shifting from coal to natural gas greatly outweigh the slightly larger upstream emissions for Natural Gas. The authors can consider to add a figure with the future trajectory of China's total GHG emissions per year with the current CO₂ intensity of NG and the effect of taking into account the future developments in supply.

Response: Thanks very much for the suggestion. We have accepted the reviewer's suggestion and added a figure with the projections of the GHG emissions reduction benefits of China's coal-to-gas switching under different scenarios of GHG intensities of natural gas supply. And thus we demonstrated the importance of analyzing and reducing the upstream emissions of gas supply for global warming mitigations.

The section of Projected GHG emissions for 2020 and 2030 in the original manuscript was revised to:

"Policy implications through 2030"

The supply-energy-weighted average GHG intensity of 2030 is projected to be 23.3 g CO₂eq MJ⁻¹ (see Fig. 4 for the GHG intensity supply curve of 2030). The increasing average GHG intensity of 2030 is caused by the potential growth of GHG-intensive gas supplies, including supplies from Russia's Urengoi and Nadym fields, Turkmenistan's Galkynysh and Bagtiyarlyk, and domestic shale gas from Fulling field, which all have well-to-city-gate GHG intensities higher than the 75-percentile level at the supply curve. High well-to-city-gate emissions of gas supply would significantly offset the potential climate benefit of China's coal-to-gas switching. Assuming the increase of gas supply in 2016~2030 would replace coal for power generation in China, with an average level of well-to-city-gate GHG intensity, the total GHG emissions reduction benefits are estimated to be 7.4 and 7.8 gigatonne CO₂eq in GWP₁₀₀ and GWP₂₀, respectively. High GHG intensity at the 80-percentile level would reduce the climate benefits of the coal-to-gas switching by 1.3 and 2.8 gigatonne CO₂eq in GWP₁₀₀ and GWP₂₀, respectively. While low GHG intensity at the 20-percentile level would increase the average climate benefits by 12% and 34% in GWP₁₀₀ and GWP₂₀, respectively (see Fig.5). The variability of well-to-city-gate GHG intensity of gas supplies and its significant effects on climate benefits demonstrates the enormous potentials of further emissions reductions through effective gas resources management and supply chain optimizations. For instance, optimizations of pipeline networks to

reduce transmission distances will help reduce emissions and costs. Monitoring and regulating methane leakages of transmission systems can achieve emissions reductions and safety improvements. Applications of CCS to the production of gas fields with exceptionally high CO₂ content can effectively reduce GHG emissions.

Fig.4 Well-to-city-gate GHG intensity supply curve of natural gas for China in 2030. *a.* GHG intensity supply curve with emission breakdowns for individual processes. Bars with numbers in parentheses on top are the top 10 gas fields with the largest supply in 2030. The numbers in parentheses are their corresponding gas field number in Table 1 and Fig. 1. Error bars represent the 90% CI of the estimates with Monte Carlo Simulation of uncertain parameter inputs (see details in SI 1). *b.* Empirical probability mass function and empirical cumulative probability function (CDF). Different colors in the empirical probability mass function indicate different categories of gas sources. GHG intensities in the figure are based on GWP₁₀₀. Source data are provided in a Source Data file.

Fig.5 Estimation of GHG emissions reduction benefits of China's coal-to-gas switching through 2030. Different lines in the figure show the projections of life cycle GHG emissions of China's power generation under difference scenarios: 1. No coal-to-gas switching, 2. Mean GHG intensity of gas supply: all growth of natural gas in 2016–2030 is assumed to replace coal in the power sector while maintaining the sum of power output from coal and natural gas the same as that for the no coal-to-gas switching scenario. The well-to-city-gate GHG intensity of natural gas in the scenario is assumed at the mean level at the supply curve. 3. Low GHG intensity of gas supply: assume the same volume of natural gas to replace coal as that in the mean GHG intensity of gas supply scenario. The well-to-city-gate GHG intensity of natural gas in the scenario is assumed at the 20-

percentile level. 4. High GHG intensity of gas supply: assume the same volume of natural gas to replace coal as that in the mean GHG intensity of gas supply scenario. The well-to-city-gate GHG intensity of natural gas in the scenario is assumed at the 80-percentile level. The projections of China's total electricity generation and other energy sources for power generation through 2030 are obtained from the international energy outlook 2019 by U.S. Energy Information Administration (EIA)⁶⁷. Life cycle GHG emissions intensity of coal and combustion emissions intensity of natural gas are obtained from a China-data-configured GREET[®]²³. Source data are provided in a Source Data file.

”

2) One of the strong points of this paper is the use of region-specific data with regards to gas composition and pipeline length. Unfortunately, as stated on lines 127-128, the leakage rates from pipelines are generic rather than region-specific. This is a shame because, this is both one the most critical variables and it is also a variable that is likely to show a high degree of variation across regions. I don't think the +/- 20% used in the sensitivity analysis does justice to this variation. I think the pipeline leakage rate is likely to show a much higher spatial/technological variability than that. This is a flaw that is not easily corrected without gathering additional data. Still I feel that the paper needs to address this issue in a much more comprehensive way before it can be published in Nature Communications. It is crucial to know this because much of the added value of the paper lies in reducing the uncertainty in China's future natural gas footprint. Without having a much better view on the uncertainty in pipeline leakage, it is impossible to judge to which extent the overall uncertainty is reduced.

Response: Thanks very much for the comment. In order to address the issue, we have conducted literature reviews to analyze the uncertainty of the pipeline leakage rate and applied the uncertain estimates of pipeline leakage rate together with other uncertain parameters in the Monte Carlo simulation to calculate the probability distributions of the LCA results.

A section about uncertainty analysis was added as Section 3 in SI I, and the first part of the section is about the uncertainty of the pipeline leakage rate. In the section, we summarized the uncertain estimate of pipeline leakage rates for the U.S. and Russia systems from literature and used them to calculate the pipeline leakage for gas transmission in the U.S. and Russia, respectively. We discussed the potential reasons for the variations in the pipeline leakage rate. Because there is no publicly available data on the leakage rate of China's pipeline system. We analyzed the characteristics of China's pipeline system that might be related to the leakage rate, including pipeline age, operating pressure, level of maintenance, and compared to those for the U.S. and Russia's pipeline system to obtain information for the uncertain estimations about China's pipeline leakage rate.

Line 123-129 in the main manuscript was revised to:

“The pipeline leakage rate is a key factor when calculating transmission-associated emissions, yet there is high uncertainty in the estimations of the parameter^{2,10,15,20,22,23,68}. Different measurement methods and varied characteristics of the pipeline system, such as pipeline age, level of maintenances and mitigation practices potentially result in variance of pipeline leakage rates^{2,10,15,20,68}. In the study, the uncertainty range of the

pipeline leakage rate is estimated and applied (with other uncertain inputs) in the Monte Carlo simulations to calculate the uncertainties of results (error bars in Fig. 1, 2 and 4). See Sensitivity Analysis and Uncertainty Analysis in SII for details.”

The following paragraphs were added in Section 3 Uncertainty Analysis of SII:

“3.1 Uncertainty of pipeline leakage rate

Leakage rate of pipeline transmission is a key factor to determine the transmission-associated and the overall emissions, especially for gas sourced from long-distance international pipelines. Enormous research efforts have been done on the measurements of pipeline system leakages since the 1990s¹⁻¹⁹. Yet there is still high uncertainty and debate on this issue^{6-8,20}.

To have a clear understanding of the uncertainty of pipeline leakage rate and its impacts on our final results, here, we conduct a literature review on the pipeline leakage studies and incorporate the uncertainty of the pipeline leakage rate into the Monte Carlo simulation to generate a robust result.

The majority of studies on pipeline transmission leakages were centered on the U.S. and Russia, which are the two largest natural gas producers that account for ~40% of gas production worldwide⁷. Pipeline leakage rates are also measured and reported in some European countries, such as United Kingdom, Germany and the Netherlands^{5,9}, but since China does not import natural gas from these European countries, studies in those countries will not be discussed here. There is no publicly available report on the leakage rate of China’s pipeline system.

Fig. S5 summarizes literature estimates of the pipeline leakage rate for the U.S. and Russia. Noteworthy, we have converted the percentage leakage rate of throughput to the leakage rate on per kilometer (km) of transport distance basis. The lengths of the U.S and Russian pipeline system that used for conversions are obtained from the literature. Studies of Russia’s pipeline system leaks are focused on Russia’s export corridors to Western European. The length of the export corridors is ~3000 km within the Russian border (the distance of Northern Corridor is 3075 km while the Central Corridor is 3376 km)^{2,11}. The average transmission distance of the U.S natural gas system was estimated at ~1000 km²¹⁻²⁴. The much longer transmission distance of Russia’s natural gas system leads to its higher leakage rate in terms of percentage throughput than that of the U.S. system in literature^{7,10,11}. Yet on the per km basis, pipeline leakage rates of the two systems are comparable and their uncertainty ranges overlap mostly, as shown in Fig. S5.

Fig. S5 Literature estimates of leakage rates of the U.S. and Russia pipeline system.

Data sources: Zimmerle et al. 2015¹², Logan et al. 2012¹³, Laurenzi et al. 2013¹⁴, Alvarez et al. 2018¹⁵, USEPA 2013²⁵, USEPA 2014²⁶, USEPA 2015²⁷, USEPA 2016²⁸, USEPA 2017²⁹, USEPA 2018³⁰, USEPA 2019³¹, Dedikov et al. 1999², Lelieveld et al.^{10,11}, Russia NIR 2012³², Russia NIR 2013³³, Russia NIR 2014³⁴, Russia NIR 2015³⁵, Russia NIR 2016³⁶, Rosstat 2015³⁷

The difference in the estimations of pipeline leakage rate is caused by many reasons. Firstly, varied measurement approaches would result in different estimates²⁰. There are generally two types of measurement approaches: the top-down and the bottom-up approaches. Top-down studies use aircraft, tower, remote sensing, etc. to measure the concentrations of CH₄ or other volatile organic compounds (VOCs) in the atmosphere^{7,15,20}. The atmospheric observations are then attributed to different sources, including multiple anthropogenic sources (such as waste landfill, production of ruminant animals, oil, coal and natural gas production, etc) and natural sources (such as natural wetland, natural geologic seeps, etc.)^{20,31} to estimate emissions associated with transmission leakages. The sources attribution is the key challenge for the top-down study and it will introduce significant uncertainties since assumptions about the activity levels of different sources are highly uncertain²⁰. Bottom-up studies measure emissions of sampling devices or facilities, and then multiple the emission factors by device counts or activity factors to make emissions inventories^{15,20,25-31}. According to the literature review by Brandt et al, inventories generated from bottom-up studies, such as the U.S. Environmental Protection Agency (EPA) GHG inventory (GHGI), systematically reported methane emissions lower than top-down measurements²⁰. Limited sample sizes and representatives, uncaptured “superemitters”, and incomplete statistics of activities and devices are responsible for the potential underestimates of bottom-up studies^{7,20}. Fig. S5 includes estimates from both top-down and bottom-up studies,

which should have captured the uncertainties from different measurement approaches.

Secondly, characteristics of the pipeline systems, such as pipeline age, operating pressure, level of maintenances and mitigation practices, would affect the pipeline leakage rates^{7,10,18,19,38}. Previous researches revealed that old compressor stations might lead to higher leaks, and pipeline system with higher operating pressures has the tendency to increase leaks, while well maintenance and mitigation measures can significantly reduce transmission leakages^{7,10,18,19,38}.

Table S9 compares the characteristics of China's pipeline system with those of the U.S. and Russia to look for clues about the level of China's transmission leakage rate. As shown in Fig. S5, compared to U.S. and Russia, China has a much younger pipeline system which might help reduce leaks while a relatively higher operating pressure (10 MPa for the major gas pipelines such as West-East gas pipeline and Central-Asia gas pipeline), which, on the other hand, would potentially increase leaks. Methane emissions are not regulated and reported in China and the country does not actively promote emissions mitigation practices in its pipeline system operations. In contrast, U.S. and Russia have promoted and deployed multiple methane emission mitigation measurements, such as forward line pumping, corrosion repair, replacement of high-bleed pneumatic devices, targeted inspection, etc, which are considered to help significantly reduce transmission leaks^{7,16-18}. Kiefner and Rosenfeld studied pipeline incidents (including leaking seals and corrossions that are related to gas leaks) reported to U.S. Pipeline and Hazardous Material Safety Administration (PHMSA) during 2002~2009 and found that 85% of the pipeline incidents are in irrespective of the pipeline age, with just 15% of the incidents related in some way to the pipeline age, and they concluded that periodically assessment, timely repairs with mitigation efforts can ensure aged pipeline's continued fitness for service³⁸.

After carefully comparing the characteristics of the Chinese pipeline system to those of the U.S. and Russia, we are not expecting the Chinese pipeline leakage rate to be significantly higher or lower than that of the U.S. or Russia. Literature estimates of pipeline leakage rates for the U.S. and Russia have wide uncertainty ranges, which should be large enough to capture the possible value for the Chinese pipeline leakage rate. According to Lelieveld et al, the true value of leakage rate for Russia's gas export transmission system must be lower than their upper estimates of 1.6% (~7.1E-06 kg/kg-km, the highest estimate for Russia in Fig. S5), because the upper estimation is calculated based on "worst-case assumptions" in numerous areas¹⁰. Balcombe et al believed that estimates of the leakage rate of pipeline transmission (exclude leakages from production and processing) above 1.6% are results of outdated data or flawed estimation methods⁷.

In the next section, we will apply the uncertainty estimates of Russia's pipeline leakage rate (which has a larger uncertainty range than that of U.S., as shown in Fig. S5) to that of the Chinese pipeline system and the connected international pipeline system, such as the Central Asia-China pipeline, and conduct Monte Carlo simulation for uncertainty analysis.

Table S9 Comparison of characteristics of natural gas pipeline system in different countries

	Year of Pipeline installed	Operating pressure	Maintenance and mitigation practice
China's pipeline system	2000s	6.3~12 MPa	No regulation and measurements on methane emissions
U.S pipeline system	Median at 1960s	3.5~10 MPa	Well-maintained with deployment of emissions mitigation practices
Russian gas export pipeline system	Median at 1980s	7.5 Mpa	Monitored leaks, promoted and deployed emissions mitigation practices

Data sources: ANL 2007³⁹, INGAA³⁸, Lelieveld et al^{10,11}, Balcombe et al⁷.

”

Stylistic improvements (figures):

3) The color schemes of figures 1, 2 and 4 are a bit harsh on the eye, perhaps pastel versions of the same colors can be used to improve the aesthetics.

Response: Thanks for the suggestion. We have changed the color schemes of figures 1,2 and 4 according to the reviewer's comment.

4) I also suggest to change the text direction on the vertical axes to improve the readability.

Response: Thanks for the suggestion. We have changed the text direction on the vertical axes in the figures.

5) Furthermore I would suggest to remove the box around these figures.

Response: Thanks very much for the suggestion. We have removed the box around the figures.

6) Please add a little spacing between parts A and B of figures 2 and 4.

Response: Thanks very much for the suggestion. We have added some spacing between parts A and B of figures 2 and 4 in the revised manuscript.

Other:

7) I have a strong preference for the use of SI units in the Supplemental Material and the main text, units such as Btu/hphr (Tables S2 and S7) or kg/bbl (Table S2) are unnecessarily complicated. The first one should even be dimensionless since both are measures of energy.

Response: Thanks very much for the suggestion. We have converted all British units to SI units in the main manuscript and the Supplemental Material. Please refer to the revised manuscript for details.

Reference

- 1 Harrison, M. R. *Methane emissions from the natural gas industry*. (US Environmental Protection Agency, National Risk Management Research Laboratory, 1996).
- 2 Dedikov, J. *et al.* Estimating methane releases from natural gas production and transmission in Russia. *Atmospheric Environment* **33**, 3291-3299 (1999).
- 3 Harrison, M. *et al.* Natural Gas Industry Methane Emission Factor Improvement Study Final Report: Cooperative Agreement No. XA-83376101. (2011).
- 4 Rabchuk, V., Ilkevich, N. & Kononov, Y. A study of methane leakage in the Soviet natural gas supply system. *Siberian Academy of Science, Irkutsk* (1991).
- 5 Mitchell, C., Sweet, J. & Jackson, T. A study of leakage from the UK natural gas distribution system. *Energy policy* **18**, 809-818 (1990).
- 6 Le Fevre, C. Methane Emissions: from blind spot to spotlight. (2017).
- 7 Balcombe, P., Anderson, K., Speirs, J., Brandon, N., and Hawkes A. Methane & CO2 emissions from the natural gas supply chain report. (Sustainable Gas Institute, Imperial College London, 2015);
- 8 Balcombe, P. *et al.* The natural gas supply chain: the importance of methane and carbon dioxide emissions. *ACS Sustainable Chemistry & Engineering* **5**, 3-20 (2016).
- 9 Müller-Syring, G., Große, C. & Glandien, J. Critical evaluation of default values for the GHG emissions of the natural gas supply chain. *Leipzig, Germany: DBI Gas-und Umwelttechnik GmbH* (2016).
- 10 Lelieveld, J. *et al.* Greenhouse gases: Low methane leakage from gas pipelines. *Nature* **434**, 841 (2005).
- 11 Lechtenböhmer, S. *et al.* Greenhouse gas emissions from the Russian natural gas export pipeline system. *Wuppertal/Mainz: Wuppertal Institute and Max Planck Institute* (2005).
- 12 Zimmerle, D. J. *et al.* Methane emissions from the natural gas transmission and storage system in the United States. *Environmental science & technology* **49**, 9374-9383 (2015).
- 13 Logan, J. *et al.* Natural gas and the transformation of the US energy sector: electricity. (National Renewable Energy Lab.(NREL), Golden, CO (United States), 2012);
- 14 Laurenzi, I. J. & Jersey, G. R. Life cycle greenhouse gas emissions and freshwater consumption of Marcellus shale gas. *Environmental science & technology* **47**, 4896-4903 (2013).
- 15 Alvarez, R. A. *et al.* Assessment of methane emissions from the US oil and gas supply chain. *Science* **361**, 186-188 (2018).
- 16 Ishkov, A. *et al.* Understanding methane emissions sources and viable mitigation measures in the natural gas transmission systems: Russian and US experience, in *International Gas Union Research Conference*.

- 17 Anifowose, B. & Odubela, M. Methane emissions from oil and gas transport facilities—exploring innovative ways to mitigate environmental consequences. *Journal of Cleaner Production* **92**, 121-133 (2015).
- 18 Lechtenböhmer, S. *et al.* Tapping the leakages: Methane losses, mitigation options and policy issues for Russian long distance gas transmission pipelines. *International journal of greenhouse gas control* **1**, 387-395 (2007).
- 19 Venugopal, S. The effective management of methane emissions from natural gas pipelines, in *Greenhouse Gas Control Technologies-6th International Conference*. 1293-1298 (Elsevier).
- 20 Brandt, A. R. *et al.* Methane leaks from North American natural gas systems. *Science* **343**, 733-735 (2014).
- 21 Littlefield, J. *et al.* Life cycle analysis of natural gas extraction and power generation. (National Energy Technology Laboratory, 2014); https://www.netl.doe.gov/projects/files/NaturalGasandPowerLCAModelDocumentationNG%20Report_052914.pdf
- 22 Skone, T. J. *et al.* Life cycle analysis of natural gas extraction and power generation. (National Energy Technology Laboratory 2016); https://www.netl.doe.gov/projects/files/LifeCycleAnalysisofNaturalGasExtractionandPowerGeneration_083016.pdf
- 23 Wang, M. *The Greenhouse Gases, Regulated Emissions, and Energy Use in Transportation (GREET) Model* (2018); <https://greet.es.anl.gov/>
- 24 Burnham, A. *et al.* Life-cycle greenhouse gas emissions of shale gas, natural gas, coal, and petroleum. *Environmental science & technology* **46**, 619-627 (2011).
- 25 Inventory of US greenhouse gas emissions and sinks: 1990–2011. (United States Environmental Protection Agency (USEPA), Washington DC, 2013);
- 26 Inventory of US greenhouse gas emissions and sinks: 1990–2012. (USEPA, Washington DC, U.S., 2014);
- 27 Inventory of us greenhouse gas emissions and sinks: 1990-2013. (USEPA, Washington, DC, 2015);
- 28 Inventory of US greenhouse gas emissions and sinks: 1990-2014. (USEPA, Washington, DC, 2016);
- 29 Inventory of US greenhouse gas emissions and sinks: 1990–2015. (USEPA, Washington, DC, 2017);
- 30 Inventory of US Greenhouse Gas Emissions and Sinks: 1990–2016. (USEPA, Washington, DC, 2018);
- 31 Inventory of US greenhouse gas emissions and sinks: 1990–2017. (USEPA, Washington, DC, 2019);
- 32 National inventory report: anthropogenic emissions by sources and removals by sinks 1990-2012. (Russian Federation Moscow, 2014);

- 33 National inventory report: anthropogenic emissions by sources and removals by sinks 1990-2012. (Russian Federation, Moscow, 2015);
- 34 National inventory report: anthropogenic emissions by sources and removals by sinks 1990-2014. (Russian Federation, Moscow, 2016);
- 35 National inventory report: anthropogenic emissions by sources and removals by sinks 1990-2015. (Russian Federation Moscow, 2017);
- 36 National inventory report: anthropogenic emissions by sources and removals by sinks 1990-2016. (Russian Federation, Moscow, 2018);
- 37 Bulletins Environmental Protection: Information on air quality in 2015. (Russian Federation Federal State Statistics Service (Rosstat), 2015); http://www.gks.ru/wps/wcm/connect/rosstat_main/rosstat/ru/statistics/publications/catalog/5e901c0042cb5cc99b49bf307f2fa3f8
- 38 Kiefner, J. & Rosenfeld, M. The role of pipeline age in pipeline safety. *Interstate Natural Gas Association of America (INGAA)* (2012).
- 39 Folga, S. Natural gas pipeline technology overview. (Argonne National Lab.(ANL), Argonne, IL (United States), 2007);
- 40 Simpson, D. A. in *Practical Onshore Gas Field Engineering* 513-571 (Gulf Professional Publishing, 2017).
- 41 El-Houjeiri, H., Vafi, K., Duffy, J., McNally, S., & Brandt, A. R. . *Oil Production Greenhouse Gas Emissions Estimator. OPGEE version 2.0: Computer program.* (2017); <https://eao.stanford.edu/research-areas/opgee>
- 42 Willis, J. *et al. Flare efficiency estimator and case studies.* (Iwa Publishing, 2014).
- 43 Dobrota, Đ., Lalić, B. & Komar, I. Problem of boil-off in LNG supply chain. *Transactions on maritime science* **2**, 91-100 (2013).
- 44 Głomski, P. & Michalski, R. Problems with determination of evaporation rate and properties of boil-off gas on board LNG carriers. *Journal of Polish CIMAC* **6**, 133-140 (2011).
- 45 Sedlaczek, R. Boil-Off in Large and Small Scale LNG Chains. *MS Pet Eng Dep Pet Eng Appl Geophys Nor Univ Sci Technol Trondheim* (2008).
- 46 Kwak, D.-H. *et al.* Energy-efficient design and optimization of boil-off gas (BOG) re-liquefaction process for liquefied natural gas (LNG)-fuelled ship. *Energy* **148**, 915-929 (2018).
- 47 Smith, T. *et al.* Third IMO GHG Study. (International Maritime Organization (IMO), London, United Kingdom, 2015);
- 48 Buhaug, Ø. *et al.* Second IMO GHG Study 2009. (International Maritime Organization (IMO), London, United Kingdom, 2009);
- 49 Zhang, Q., Li, Z., Wang, G. & Li, H. Study on the impacts of natural gas supply cost on gas flow and infrastructure deployment in China. *Applied energy* **162**, 1385-1398 (2016).

- 50 Rioux, B. *et al.* The economic impact of price controls on China's natural gas supply chain. *Energy Economics* **80**, 394-410 (2019).
- 51 Ishwaran, M. *et al.* in *China's Gas Development Strategies* 197-232 (Springer, 2017).
- 52 *Shanghai Petroleum and Gas Exchange* (2019); <https://www.shpgx.com/html/index.html>
- 53 Trade map: Trade statistics for international business development. (International Trade Center (ITC), 2019); <https://www.trademap.org>
- 54 *General Administration of Customs of China* (2019); <http://www.customs.gov.cn/>
- 55 Kang, Z. Natural gas supply-demand situation and prospect in China. *Natural Gas Industry B* **1**, 103-112 (2014).
- 56 Lin, B. & Wang, T. Forecasting natural gas supply in China: production peak and import trends. *Energy Policy* **49**, 225-233 (2012).
- 57 Qin, Y., Edwards, R., Tong, F. & Mauzerall, D. L. Can switching from coal to shale gas bring net carbon reductions to China? *Environmental science & technology* **51**, 2554-2562 (2017).
- 58 Chang, Y., Liu, X. & Christie, P. Emerging shale gas revolution in China. *Environmental science & technology* **46**, 12281-12282 (2012).
- 59 Development Plan of Shale Gas (2016-2020) (National Energy Administration of China, 2016); http://www.gov.cn/xinwen/2016-09/30/content_5114313.htm
- 60 Policy of shale gas industry (National Energy Administration, 2013); http://zfxgk.nea.gov.cn/auto86/201310/t20131030_1715.htm
- 61 Michael F. Farina, A. W. China's age of gas: Innovation and change for energy development. (General Electric Company, 2013);
- 62 Eggleston, S. *et al.* *2006 IPCC guidelines for national greenhouse gas inventories*. Vol. 5 (Institute for Global Environmental Strategies Hayama, Japan, 2006).
- 63 Dudley, B. BP statistical review of world energy. *BP Statistical Review, London, UK, accessed Aug 6*, 2018 (2018).
- 64 Upstream Oil & Gas. (Wood Mackenzie, 2019); <https://www.woodmac.com/our-expertise/capabilities/upstream-oil-and-gas/>
- 65 Guan, D. *et al.* The gigatonne gap in China's carbon dioxide inventories. *Nature Climate Change* **2**, 672 (2012).
- 66 Yong, G. Eco-indicators: improve China's sustainability targets. *Nature* **477**, 162 (2011).
- 67 International Energy Outlook 2019. (U.S. Energy Information Administration (EIA), 2019); <https://www.eia.gov/outlooks/ieo/>
- 68 Inventory of US greenhouse gas emissions and sinks 1990-2017 (Office of Policy US Environmental Protection Agency, Planning, and Evaluation, 2018); <https://www.epa.gov/sites/production/files/2019-04/documents/us-ghg-inventory-2019-main-text.pdf>

REVIEWERS' COMMENTS:

Reviewer #1 (Remarks to the Author):

The authors have made a serious attempt to address the suggestions for revision made by Reviewer #1. I believe that the paper has been significantly improved following the response of the authors to the suggestions made by both reviewers, and should now be accepted for publication.

Reviewer #2 (Remarks to the Author):

I feel the authors have addressed my original comments in an appropriate manner.

Manuscript ID: NCOMMS-19-28963-B

Carbon footprint of global natural gas supplies to China

**Yu Gan, Hassan El-Houjeiri, Alhassan Badahdah, Zifeng Lu, Hao Cai, Steven Przesmitzki,
Michael Wang**

Reviewers' comments:

Reviewer #1 (Remarks to the Author):

The authors have made a serious attempt to address the suggestions for revision made by Reviewer #1. I believe that the paper has been significantly improved following the response of the authors to the suggestions made by both reviewers, and should now be accepted for publication.

Reviewer #2 (Remarks to the Author):

I feel the authors have addressed my original comments in an appropriate manner.

Responds: We sincerely thank the reviewers for their time and comments. There are no questions that need to be addressed.